# Eligibility Traces for Confounding Robust Off-Policy Evaluation: a Causal Approach

## Abstract

A unifying theme in Artificial Intelligence is learning an effective policy to control an agent in an unknown environment in order to optimize a certain performance measure. Off-policy methods can significantly improve the sample efficiency during training since they allow an agent to learn from observed trajectories generated by different *behavior policies*, without directly deploying the *target policies* in the underlying environment. This paper studies off-policy evaluation from biased offline data where (1) *unobserved confounding* bias cannot be ruled out a priori; or (2) the observed trajectories do not *overlap* with intended behaviors of the learner, i.e., the target and behavior policies do not share a common support. Specifically, we first extend the Bellman's equation to derive effective closed-form bounds over value functions from the observational distribution contaminated with unobserved confounding and no-overlap. Second, we propose two novel algorithms that use eligibility traces to estimate these bounds from finite observational data. Compared to other partial identification methods for off-policy evaluation in sequential environments, these methods are model-free and do not rely on additional parametric knowledge about the system dynamics in the underlying environment.

## 1 Introduction

A typical reinforcement learning agent learns from past data, i.e., from observed trajectories of states, actions, and reward signals generated by the agent intervening in the underlying environment. This data reflects the influence of the decision-making policy used to allocate actions based on the observed state, which is called the *behavior policy*. This policy might be selected by the agent in the past or by a different demonstrator operating in the same environment. *Policy evaluation* studies the problem of evaluating the effectiveness of a candidate *target policy* from the combination of past data and theoretical assumptions about the environment. When the behavior and target policies coincide, the evaluation is called *on-policy* learning, in which the expected return of candidate policies given the agent's starting state (i.e., the value function) could be directly estimated with empirical means (Sutton & Barto, 1998). In practice, however, the learner might have to learn about policies different from the currently deployed one that generated the data, leading to the *off-policy* learning problem.

Off-policy learning is a popular area of research, as it allows for more efficient learning by using data from different policies. Several algorithms have been proposed for off-policy evaluation from finite observations, including Q-learning (Watkins, 1989; Watkins & Dayan, 1992), importance sampling (Swaminathan & Joachims, 2015; Jiang & Li, 2016), and temporal difference (Precup et al., 2000; Munos et al., 2016). These algorithms rely on two critical assumptions about the behavior policy. First, no unobserved confounder affects the behavior policy's selected action and the subsequent state and reward. Second, the behavior policy is stochastic, covering all intended actions the target policy selects given all observed states. When either of these assumptions does not hold, the effect of the target policy is generally not *identifiable*, i.e., the model assumptions are insufficient to uniquely determine the value function from the offline data (Pearl, 2000; Zhang & Bareinboim, 2019).

In recent times, researchers have been using partial identification methods to obtain reliable off-policy evaluation in situations where there are unobserved confounders, and the behavior and target policies have no common support (Kallus & Zhou, 2018; Zhang & Bareinboim, 2019; Kallus & Zhou, 2020; Namkoong et al., 2020; Khan et al., 2023; Bruns-Smith & Zhou, 2023; Kausik et al., 2024). Partial identification is a well-studied problem in causal inference (Balke & Pearl, 1997; Zhang et al., 2022),

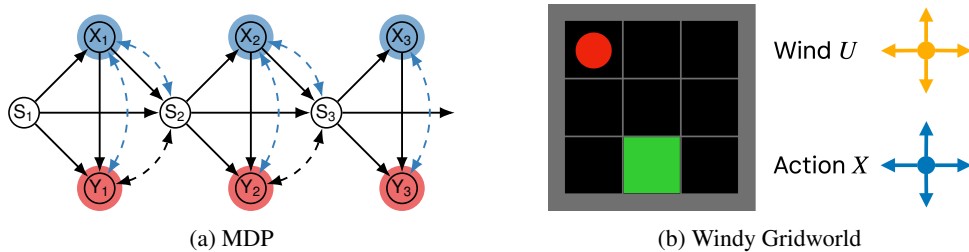

(a) MDP                          (b) Windy Gridworld

Figure 1: (a) Causal diagram representing the data-generating mechanisms in a Markov Decision Process (MDP); (b) A windy gridworld environment where the red dot represents the agent and green square is the goal state; the agent can take five actions - `up`, `down`, `right`, `left`, and `stay-put`; the wind can blow in five direction - `north`, `south`, `west`, `east`, and `no-wind`.

econometrics (Imbens & Rubin, 1997; Poirier, 1998; Romano & Shaikh, 2008; Stoye, 2009; Bugni, 2010; Todem et al., 2010; Moon & Schorfheide, 2012), and dynamical systems (Bajari et al., 2007; Norets & Tang, 2014; Dickstein & Morales, 2018; Morales et al., 2019; Berry & Compiani, 2023). It enables the derivation of informative bounds on target effects from confounded observational data. Among these works, researchers often employ a combination of approaches and constraints. These include (1) the marginal sensitivity model presuming access to a bound over the odds ratio between the nominal and actual behavioral policies (Kallus & Zhou, 2018; 2020; Namkoong et al., 2020; Khan et al., 2023; Bruns-Smith & Zhou, 2023); (2) additional parametric assumptions about the system dynamics (i.e., reward function and transition distribution) are invoked under which bounds are derived (Khan et al., 2023; Kausik et al., 2024); (3) a model-based algorithm is applied, which requires estimation of the underlying system dynamics (Zhang & Bareinboim, 2019); (4) the decision horizon is finite, i.e., the agent only determines a finite number of actions (Kallus & Zhou, 2018; Zhang & Bareinboim, 2019; Namkoong et al., 2020; Khan et al., 2023; Kausik et al., 2024). See Appendix A for a more detailed survey on partial identification and robust reinforcement learning.

This paper studies model-free algorithms for robust off-policy evaluation from confounded offline data generated by behavior policy with no-overlap support. We propose novel partial identification algorithms using eligibility traces to obtain informative bounds over the expected return of candidate policies from offline data generated from an unknown Markov decision process with an infinite horizon. More specifically, our contributions are summarized as follows. (1) We extend the Bellman equation that permits one to derive optimal bounds over target value functions from the observational distribution generated by an unknown behavior policy. (2) We propose a causal off-policy temporal difference algorithm (`C-TD(λ)`) using eligibility traces to estimate bounds over the state value function from finite observations contaminated with unobserved confounding and no-overlap. (3) We introduce an alternative eligibility traces algorithm following tree backup (`C-TB(λ)`) that obtains bounds over the state-action value function from confounded observations. Finally, we evaluate our proposed algorithms using extensive simulations in synthetic environments. Due to space constraints, all proofs are provided in Appendix B; details on the experiment setup are provided in Appendix C.

**Preliminaries and Notations** We use capital letters to denote random variables ($X$), small letters for their values ($x$) and $\mathscr{D}_X$ for the domain of $X$. For an arbitrary set $\boldsymbol{X}$, let $|\boldsymbol{X}|$ be its cardinality. Fix indices $i, j \in \mathbb{N}$. Let $\boldsymbol{X}_{i:j}$ stand for a sequence of variables $\{X_i, X_{i+1}, \ldots, X_j\}$; $\boldsymbol{X}_{i:j} = \emptyset$ if $j < i$. We denote by $P(\boldsymbol{X})$ represents a probability distribution over variables $\boldsymbol{X}$. Similarly, $P(\boldsymbol{Y} \mid \boldsymbol{X})$ represents a set of conditional distributions $P(\boldsymbol{Y} \mid \boldsymbol{X} = \boldsymbol{x})$ for all realizations $\boldsymbol{x}$. We will consistently use $P(\boldsymbol{x})$ as abbreviations for probabilities $P(\boldsymbol{X} = \boldsymbol{x})$; so does $P(\boldsymbol{Y} = \boldsymbol{y} \mid \boldsymbol{X} = \boldsymbol{x}) = P(\boldsymbol{y} \mid \boldsymbol{x})$. Finally, $\mathbb{1}_{\boldsymbol{Z}=\boldsymbol{z}}$ is an indicator function that returns 1 if event $\boldsymbol{Z} = \boldsymbol{z}$ holds true; otherwise, it returns 0.

An SCM $M$ is a tuple $\langle \boldsymbol{V}, \boldsymbol{U}, \mathcal{F}, P(\boldsymbol{U}) \rangle$, where $\boldsymbol{V}$ is a set of endogenous variables and $\boldsymbol{U}$ is a set of exogenous variables (Pearl, 2000; Bareinboim et al., 2022). $\mathcal{F}$ is a set of functions s.t. each $f_V \in \mathcal{F}$ decides values of an endogenous variable $V \in \boldsymbol{V}$ taking as argument a combination of other variables in the system. That is, $V \leftarrow f_V(\boldsymbol{PA}_V, \boldsymbol{U}_V), \boldsymbol{PA}_V \subseteq \boldsymbol{V}, \boldsymbol{U}_V \subseteq \boldsymbol{U}$. Values of exogenous variables $U \in \boldsymbol{U}$ are drawn from the exogenous distribution $P(\boldsymbol{U})$. Naturally, $M$ induces an *observational distribution* $P(\boldsymbol{V})$. An intervention on a subset $\boldsymbol{X} \subseteq \boldsymbol{V}$, denoted by $\mathrm{do}(\boldsymbol{x})$, is an operation where values of $\boldsymbol{X}$ are set to constants $\boldsymbol{x}$, replacing the functions $\{f_X : \forall X \in \boldsymbol{X}\}$ that would normally determine their values. For an SCM $M$, let $M_{\boldsymbol{x}}$ be a submodel of $M$ induced by intervention $\mathrm{do}(\boldsymbol{x})$.

For a set $\boldsymbol{Y} \subseteq \boldsymbol{V}$, the *interventional distribution* $P_{\boldsymbol{x}}(\boldsymbol{Y})$ induced by do($\boldsymbol{x}$) is defined as the joint distribution over $\boldsymbol{Y}$ in the submodel $M_{\boldsymbol{x}}$, i.e., $P_{\boldsymbol{x}}(\boldsymbol{Y}; M) \triangleq P(\boldsymbol{Y}; M_{\boldsymbol{x}})$.

## 2 CHALLENGES OF CAUSAL INCONSISTENCY

We will focus on the policy evaluation problem of an agent operating in a Markov Decision Process (MDP) (Puterman, 1994) over a series of interventions $t = 1, 2, \ldots$. For every time step $t$, the agent observes the current state $S_t$, performs an action do($X_t$), receives a subsequent reward $Y_t$, and moves to the next state $S_{t+1}$. Values of the action $X_t$ are selected by sampling from a stationary policy $\pi(x \mid s)$, which is a function mapping from the domain of the observed state $S_t$ to the probability space over the domain of action $X_t$. Let $\boldsymbol{U}_t$ be an unobserved noise independently drawn from an exogenous distribution $P(\boldsymbol{U})$. Values of the reward $Y_t$ and the next state $S_{t+1}$ are, respectively, determined by structural functions $y_t \leftarrow f_Y(s_t, x_t, \boldsymbol{u}_t)$ and $s_{t+1} \leftarrow f_S(s_t, x_t, \boldsymbol{u}_t)$, taking as input the current state $S_t$, action $X_t$, and latent noise $\boldsymbol{U}_t$; values of $S_1$ are drawn from an initial distribution $P(S_1)$. We will consistently use $\mathcal{X}$, $\mathcal{S}$, and $\mathcal{Y}$ to denote the domain of every action $X_t$, state $S_t$, and reward $Y_t$. Like a standard discrete MDP, domains of actions $\mathcal{X}$ and states $\mathcal{S}$ are assumed to be finite; rewards are bounded in a real interval $\mathcal{Y} \triangleq [a, b] \subset \mathbb{R}$. Naturally, the agent operating in this environment defines an interventional distribution $P_\pi$ summarizing the consequences of its actions.

Fig. 1a shows a graphical representation (for now, without the highlighted bi-directed arrows) of this data-generating process where nodes represent observed variables and directed arrows represent the functional relationships between them. For every time step $t > 1$, the current state $S_t$ "block" all pathways from previous nodes (e.g., $S_{t-1}$) to the future nodes (e.g., $S_{t+1}$) (Pearl, 2000, Def. 1.2.3). Applying the d-separation rules leads to the following independence relationships in distribution $P_\pi$.

**Definition 1** (Markov Property (Puterman, 1994)). For a distribution $P_*$ over a sequence of states $S_1, S_2, \ldots$, actions $X_1, X_2, \ldots$, and rewards $Y_1, Y_2, \ldots$, the Markov property holds if for every $t = 1, 2, \ldots, \left( \bar{\boldsymbol{S}}_{1:t-1}, \bar{\boldsymbol{X}}_{1:t-1}, \bar{\boldsymbol{Y}}_{1:t-1} \perp\!\!\!\perp \bar{\boldsymbol{X}}_{t:\infty}, \bar{\boldsymbol{S}}_{t+1:\infty}, \bar{\boldsymbol{Y}}_{t:\infty} \mid S_t \right)$ with regard to distribution $P_*$.

It follows from Def. 1 that for any horizon $T$, the distribution generated by a policy $\pi$ factories as

$$P_\pi(\bar{\boldsymbol{x}}_{1:T}, \bar{\boldsymbol{s}}_{1:T}, \bar{\boldsymbol{y}}_{1:T}) = P(s_1) \prod_{t=1}^{T} \pi(x_t \mid s_t) \mathcal{T}(s_t, x_t, s_{t+1}) \mathcal{R}(s_t, x_t, y_t) \tag{1}$$

where the transition distribution $\mathcal{T}$ and the reward distribution $\mathcal{R}$ are interventional queries given by

$$\mathcal{T}(s_t, x_t, s_{t+1}) = P_{x_t}(s_{t+1} \mid s_t) = \int_{\boldsymbol{u}_t} \mathbb{1}_{s_{t+1} = f_S(s_t, x_t, \boldsymbol{u}_t)} P(\boldsymbol{u}_t) \tag{2}$$

$$\mathcal{R}(s_t, x_t, y_t) = P_{x_t}(y_t \mid s_t) = \int_{\boldsymbol{u}_t} \mathbb{1}_{y = f_Y(s_t, x_t, \boldsymbol{u}_t)} P(\boldsymbol{u}_t) \tag{3}$$

For convenience, we write the reward function $\mathcal{R}(s, x)$ as the expected value $\sum_y y \mathcal{R}(s, x, y)$. Fix a discounted factor $\gamma \in [0, 1]$. A common objective for an agent is to optimize its cumulative return $R_t = \sum_{i=0}^{\infty} \gamma^i Y_{t+i}$. In analysis, we often evaluate the state value function $V_\pi(s)$, which is the expected return given the agent's starting state $S_t = s$. That is, $V_\pi(s) = \mathbb{E}_\pi[R_t \mid S_t = s]$. A similar state-action value function $Q_\pi(s, x)$ is defined as the expected return starting from state $s$, taking action $x$ and thereafter following policy $\pi$, i.e., $Q_\pi(s, x) = \mathbb{E}_{X_t \leftarrow x, \pi}[R_t \mid S_t = s]$. One could recursively evaluate the value function of any state $s$ using the *Bellman Equation* (Bellman, 1966):

$$V_\pi(s) = \sum_x \pi(x \mid s) \left( \mathcal{R}(s, x) + \gamma \sum_{s'} \mathcal{T}(s, x, s') V_\pi(s') \right) \tag{4}$$

Similarly, an analogous equation for the state-action value function is

$$Q_\pi(s, x) = \mathcal{R}(s, x) + \gamma \sum_{s'} \mathcal{T}(s, x, s') V_\pi(s') \tag{5}$$

**Off-Policy Evaluation** Despite the effectiveness of planning algorithms, they require detailed parametrization of the transition distribution $\mathcal{T}$ and the reward function $\mathcal{R}$, which are not accessible in many real-world applications. This means that a learning process must take place. A common

approach is off-policy learning, where the agent has access to observed trajectories generated by a *behavioral policy* $f_X$, different from the target policy $\pi$, operating in the same environment. More specifically, for every time step $t$, the behavioral policy selects an action $X_t \leftarrow f_X(s_t, \boldsymbol{u}_t)$ based on the current state $S_t = s_t$ and latent noise $\boldsymbol{U}_t = \boldsymbol{u}_t$. Fig. 1a shows the graphical representation of the data-generating process of the behavior policy; the added bi-directed arrows, e.g., $X_t \longleftrightarrow Y_t$, indicate the presence of an unobserved confounder $U \in \boldsymbol{U}_t$ affecting both the action $X_t$ and outcome $Y_t$. We summarize observed trajectories of the behavior policy using the observational distribution $P$.

Off-policy evaluation attempts to estimate the effects of a candidate policy $\pi(x|s)$ from the observational data generated by the behavior policy $f_X$. Standard off-policy methods focus on the identifiable setting where the target transition distribution $\mathcal{T}$ and reward function $\mathcal{R}$ remain consistent in both the interventional $P_\pi$ and observational distribution $P$. Formally,

**Definition 2** (Causal Consistency). For an interventional distribution $P_\pi$ and an observational distribution $P$ satisfying the Markov property (Def. 1), the Causal Consistency holds with regard to $P_\pi$ and $P$ if the following statement holds, for every time step $t = 1, 2, \ldots$,

$$P_{x_t}(s_{t+1} \mid s_t) = P(s_{t+1} \mid s_t, x_t), \quad \text{and} \quad P_{x_t}(y_t \mid s_t) = P(y_t \mid s_t, x_t) \tag{6}$$

When Def. 2 holds, the learner could recover the parametrization of the transition distribution $\mathcal{T}$ and reward function $\mathcal{R}$ from the observational data, following the identification formula in Eq. (6). Several off-policy algorithms have been proposed to estimate the effect of candidate policies from finite observations under causal consistency (Watkins, 1989; Watkins & Dayan, 1992; Swaminathan & Joachims, 2015; Jiang & Li, 2016; Precup et al., 2000; Munos et al., 2016).

There exist graphical criteria in the literature (Pearl & Robins, 1995; Shpitser et al., 2010; Perković et al., 2015) to evaluate whether causal consistency (Def. 2) holds from causal knowledge of the environment, including the celebrated *backdoor* criterion (Pearl, 2000, Def. 3.3.1). However, in many practical applications, causal consistency could be fragile and does not necessarily hold due to some violations in the generative process. These include: (1) there exists an unobserved confounder affecting the action $X_t$ and subsequent outcomes $Y_t$, $S_{t+1}$ simultaneously (blue, dashed arrows in Fig. 1a); (2) there is no overlap in the support between the target and behavior policies, i.e., the propensity score $P(x_t \mid s_t) = 0$ for some state-action pair $s_t, x_t$. When either of these violations occurs, applying standard off-policy methods may fail to recover the expected return of the target policy, leading to estimation bias. The following example illustrates such challenges.

**Example 1** (Windy Gridworld). Consider a Windy Gridworld described in Fig. 1b, where the red dot represents the agent and the green square represents the goal state. The agent can take five actions $X_t$ - up, down, right, left, and stay-put. However, the agent's movement is affected by the wind; the direction of the wind $U_t$ includes - north, south, west, east, and no-wind. For every time step, the agent receives a constant reward $Y_t \leftarrow -1$. The next state of the agent is shifted by both its action and the wind direction through the mechanism $S_{t+1} \leftarrow S_t + X_t + U_t$.

Our goal is to evaluate the expected return of a target policy $\pi^*$ described in Fig. 2a, which consistently moves towards the goal state regardless of the wind direction. As an input, we have access to observed trajectories generated by a behavior policy $X_t \leftarrow f_X(S_t, U_t)$, which could sense the wind and select an action accordingly. For example, when the agent is located in the top-left corner ($S_t = (0,0)$) and the wind is blowing south ($U_t = (0,1)$), the behavior policy will decide to move right ($X_t = (1,0)$) so that the agent could get close to the center ($S_{t+1} = (1,1)$).

Figs. 2b to 2d shows the value function estimation obtained by standard off-policy methods, including Q-Learning, one-step Temporal Difference (TD), and Eligibility Traces (TD($\lambda$)). We also include in Fig. 2e the ground truth value function computed from the underlying model parameters. The simulation reveals that standard off-policy evaluation deviates from the ground truth return. In this observational data, the wind direction $U_t$ is thus an unobserved confounder affecting both the action $X_t$ and next state $S_{t+1}$, violating causal consistency. See Appendix C for additional discussions.

## 2.1 PARTIAL CAUSAL IDENTIFICATION IN MDPS

For the remainder of this section, we will introduce partial identification methods for off-policy evaluation that is robust to the unobserved confounding and no-overlap. For every time step $t = 1, 2, \ldots$, let the reward $Y_t$ be bounded in a real interval $[a, b]$. By applying a similar bounding strategy

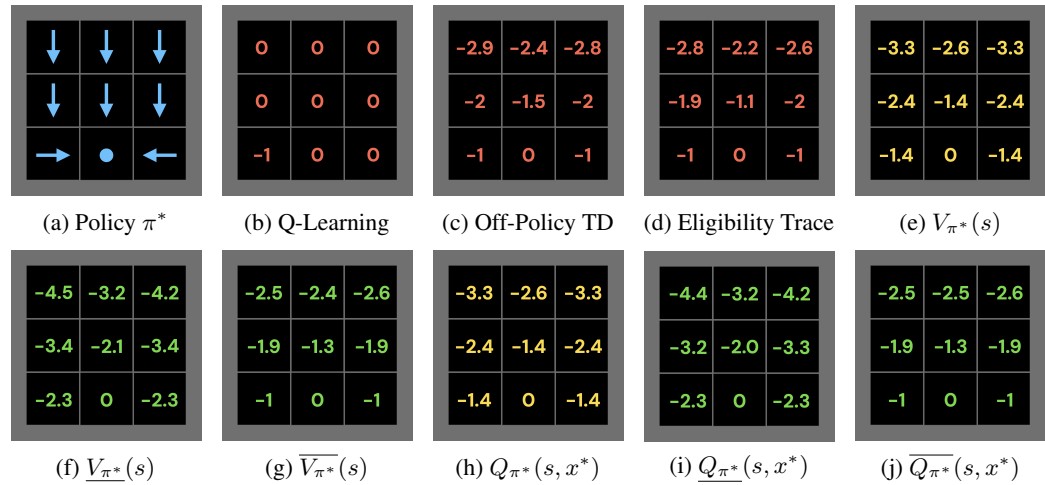

(a) Policy $\pi^*$    (b) Q-Learning    (c) Off-Policy TD    (d) Eligibility Trace    (e) $V_{\pi^*}(s)$

(f) $\underline{V_{\pi^*}}(s)$    (g) $\overline{V_{\pi^*}}(s)$    (h) $Q_{\pi^*}(s, x^*)$    (i) $\underline{Q_{\pi^*}}(s, x^*)$    (j) $\overline{Q_{\pi^*}}(s, x^*)$

Figure 2: (a) The target policy $\pi^*$ selecting an action based on the agent's location. (b - d) Value function estimation was obtained by standard off-policy methods. (e - g) The ground-truth state value function computed from the model parametrization and its lower and upper bounds estimated using the extended Bellman equation in Thm. 1. (h - j) The ground-truth state-action value function computed from the model parametrization for actions $x^* \leftarrow \pi^*(s)$ selected by the target policy and its lower and upper bounds computed from the extended Bellman equation in Thm. 2

in (Manski, 1990; Zhang & Bareinboim, 2019; Joshi et al., 2024), we derive the following bounds over the transition distribution $\mathcal{T}$ and reward function $\mathcal{R}$, for every realization $(s, x, s') \in \mathcal{S} \times \mathcal{X} \times \mathcal{S}$,

$$\mathcal{T}(s, x, s') \in \left[ \widetilde{\mathcal{T}}(s, x, s') P(x \mid s), \widetilde{\mathcal{T}}(s, x, s') P(x \mid s) + P(\neg x \mid s) \right] \quad (7)$$

$$\mathcal{R}(s, x) \in \left[ \widetilde{\mathcal{R}}(s, x) P(x \mid s) + a P(\neg x \mid s), \widetilde{\mathcal{R}}(s, x) P(x \mid s) + b P(\neg x \mid s) \right] \quad (8)$$

Among the above quantities, $P(x \mid s)$ stands for the propensity score $P(X_t = x \mid S_t = s)$ and $P(\neg x \mid s) = 1 - P(x \mid s)$; $\widetilde{\mathcal{T}}$ and $\widetilde{\mathcal{R}}$ are the nominal transition distribution and reward function computed from the observational distribution as follows:

$$\widetilde{\mathcal{T}}(s, x, s') = P(S_{t+1} = s' \mid S_t = s, X_t = x), \qquad \widetilde{\mathcal{R}}(s, x) = \mathbb{E}[Y_t \mid S_t = s, X_t = x] \quad (9)$$

In order to bound the value function $V_\pi(s)$ at state $s$ induced by a candidate policy $\pi$, one could minimize/maximize the optimization program using the Bellman's equation in Eq. (4) as the objective function, subject to constraints in Eqs. (7) and (8). Interestingly, this optimization problem is equivalent to a linear program; solving it leads to the following *extended Bellman equation*.

**Theorem 1** (Causal Bellman Equation). *For an MDP environment $M$ with reward $Y_t \in [a, b] \subseteq \mathbb{R}$, for any policy $\pi(x \mid s)$, its state value function $V_\pi(s) \in \left[ \underline{V_\pi}(s), \overline{V_\pi}(s) \right]$ for every state $s \in \mathcal{S}$, where bounds $\underline{V_\pi}, \overline{V_\pi}$ are solutions given by the following dynamic programs,*[1]

$$\left\langle \underline{V_\pi}(s), \overline{V_\pi}(s) \right\rangle = \sum_x P(x \mid s) \left( \pi(x \mid s) \left( \widetilde{\mathcal{R}}(s, x) + \gamma \sum_{s', x'} \widetilde{\mathcal{T}}(s, x, s') \left\langle \underline{V_\pi}(s'), \overline{V_\pi}(s') \right\rangle \right) \right. \quad (10)$$

$$\left. + \pi(\neg x \mid s) \left( \langle a, b \rangle + \gamma \left\langle \min_{s'} \underline{V_\pi}(s'), \max_{s'} \overline{V_\pi}(s') \right\rangle \right) \right) \quad (11)$$

Thm. 1 can be seen as an extension of the Bellman equation using the confounded observational distribution with no-overlap. For instance, in the lower bound $\underline{V_\pi}(s)$, Eq. (10) follows the standard iterative step in Bellman equation in Eq. (4), measuring the expected return when the target policy's

---

[1] $\langle a, b \rangle$ is a vector containing a lower bound $a$ and an upper bound $b$. We highlight quantities that are different from the standard Bellmen Equation.

action coincides with the observed action selected by the behavior policy; Eq. (11) could be thought as a regularizing term measuring the uncertainty due to unobserved confounding. Finally, both terms are weighted by the nominal propensity score $P(x \mid s) = P(X_t = x \mid S_t = s)$. The same derivation also applies to the upper bound $\overline{V}_\pi(s)$. An analogous extended Bellman equation bounding the state-action value function from the observational distribution can also be derived as follows.

**Theorem 2** (Causal Bellman Equation). *For an MDP environment $M$ with reward signals $Y_t \in [a, b] \subseteq \mathbb{R}$, for any policy $\pi(x \mid s)$, its state-action value function $Q_\pi \in \left[\underline{Q_\pi}(s,x), \overline{Q_\pi}(s,x)\right]$ for any state-action pair $(s, x) \in \mathcal{S} \times \mathcal{X}$, where bounds $\underline{Q_\pi}, \overline{Q_\pi}$ are given by as follows,*

$$\left\langle \underline{Q_\pi}(s,x), \overline{Q_\pi}(s,x) \right\rangle = P(x \mid s)\left( \widetilde{\mathcal{R}}(s,x) + \gamma \sum_{s',x'} \widetilde{\mathcal{T}}(s,x,s') \left\langle \underline{V_\pi}(s'), \overline{V}_\pi(s') \right\rangle \right) \quad (12)$$

$$+ P(\neg x \mid s)\left( \langle a, b \rangle + \gamma \left\langle \min_{s'} \underline{V_\pi}(s'), \max_{s'} \overline{V}_\pi(s') \right\rangle \right) \quad (13)$$

Among the bounds in Thm. 2, Eq. (12) is the standard iterative step of the Bellman equation in Eq. (5), weighted by the score $P(x \mid s)$. It estimates the expected return of performing action $do(x)$ at state $s$ when such action matches the one selected by the behavior policy. Eq. (13) is a regularized term accounting for uncertainties when the intervention $do(x)$ is not observed in the offline data. Since Thms. 1 and 2 are closed-form solutions of optimization programs and the observational constraints in Eqs. (7) and (8) are tight, the extended Bellman's equation bounds are sharp from offline data and Markov property. This means they cannot be improved without additional assumptions.

**Example 2** (Windy Gridworld Continued). Consider again the Windy Gridworld described in Example 1. We compute the lower and upper bounds over the state value function following the extended Bellman equation in Thm. 1, and provide them in Figs. 2f to 2g. We also include in Fig. 2h the ground truth state-action value function for the action $x^* \leftarrow \pi^*(s)$ selected by the target policy. The corresponding lower and upper bounds are shown in Figs. 2i to 2j, following the algorithmic procedure described in Thm. 2. The analysis reveals that the derived bounds are consistent with the ground truth value functions, corroborating the sufficiency of our proposed approach.

## 3 CONFOUNDING ROBUST ELIGIBILITY TRACES

The extended Bellman equations described so far require one to have precise estimations for the full models of the nominal transition distribution $\mathcal{T}_{\text{obs}}$, reward function $\mathcal{R}_{\text{obs}}$, and the propensity score $P(x \mid s)$. This section will introduce novel model-free algorithms, using eligibility traces (Sutton, 1988), to bound value functions from finite observational samples.

We consider the episodic framework, where the agent interacts with the environment for repeated episodes $n = 1, 2, 3, \ldots$; each episode contains a finite number of time steps $t = 1, 2, \ldots, T_n$. At each episode, the environment starts at state $s_1$ following the initial distribution $P(S_1)$. At each time step $t$, taking the observed state $s_t$ of the environment as input, the behavior policy selects an action $x_t$. In response to intervention $do(x_t)$, the environment produces a subsequent reward $y_t$ and moves to the next observed state $s_{t+1}$. If the next state $s_{t+1}$ is *terminal*, the episode terminates at time step $T_n = t + 1$; the learner receives observational data $\{\bar{x}_{1:T_n-1}, \bar{s}_{1:T_n}, \bar{y}_{1:T_n-1}\}$.

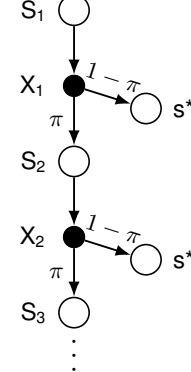

Figure 3: Backup diagram for `C-TD`$(\lambda)$.

### 3.1 CAUSAL TEMPORAL DIFFERENCE

We first introduce a novel augmentation procedure on the celebrated temporal difference (`TD`, (Sutton, 1988; Precup et al., 2000)) that allows one to estimate the bounds over state value functions, which we call the *causal temporal difference* (`C-TD`). Fig. 3 shows the backup diagram illustrating the idea of our proposed algorithm. Similar to the standard off-policy `TD`, our algorithm will update the estimation of state value functions $\underline{V_\pi}, \overline{V}_\pi$ using the sampled trajectories of transitions in the observational data. It could use a finite number of $n$-step trajectories or the entire trajectory. Different from the standard off-policy `TD`, our proposed algorithm does not weight each step of the transition using importance sampling (or equivalently, inverse propensity weighting) since the true behavior policy $f_X$ (propensity

---

**Algorithm 1** Causal Temporal Difference (C-TD($\lambda$))

---

**Require:** Observational data $\mathcal{D}$ and a candidate policy $\pi(x \mid s)$.
  1: Update the eligibility traces for all state $s$,

$$e_t(s) = \begin{cases} \gamma\lambda\pi(x_{t-1} \mid s_{t-1})e_{t-1}(s) & \text{if } s \neq s_t \\ \gamma\lambda\pi(x_{t-1} \mid s_{t-1})e_{t-1}(s) + 1 & \text{if } s = s_t \end{cases} \tag{15}$$

  where $\lambda \in [0,1]$ is an eligibility trace decay factor.
  2: Compute the temporal difference error

$$\delta_t = \pi(x_t \mid s_t)\left(y_t + \gamma V_t(s_{t+1})\right) + \pi(\neg x_t \mid s_t)\left(w + \gamma V_t(s^*)\right) - V_t(s_t) \tag{16}$$

  3: Update the value function $V_{t+1}(s) \leftarrow V_t(s) + \alpha e_t(s)\delta_t$ for all state $s$.

---

**Algorithm 2** Causal Tree-Backup (C-TB($\lambda$))

---

**Require:** Observational data $\mathcal{D}$ and a candidate policy $\pi(x|s)$.
  1: Update the eligibility traces for all state-action pairs $s, x$,

$$e_t(s,x) = \begin{cases} \gamma\lambda\pi(x_t \mid s_t)\mathbb{1}_{x_{t-1}=x}e_{t-1}(s,x) & \text{if } s \neq s_t \\ \gamma\lambda\pi(x_t \mid s_t)\mathbb{1}_{x_{t-1}=x}e_{t-1}(s,x) + 1 & \text{if } s = s_t \end{cases} \tag{17}$$

  where $\lambda \in [0,1]$ is an eligibility trace decay factor.
  2: Compute the temporal difference error for every action $x$

$$\delta_t(x) = \begin{cases} y_t + \gamma \sum_{x'} \pi(x \mid s_{t+1})Q_t(s_{t+1}, x') - Q_t(s_t, x) & \text{if } x = x_t \\ w + \gamma \sum_{x'} \pi(x' \mid s^*)Q_t(s^*, x') - Q_t(s_t, x) & \text{if } x \neq x_t \end{cases} \tag{18}$$

  3: Update the action-value function $Q_{t+1}(s,x) \leftarrow Q_t(s,x) + \alpha e_t(s,x)\delta_t(x)$ for all $s, x$.

---

score) is not recoverable from the observational data. Instead, C-TD weights each transition using the target policy $\pi$ and adjusts for the misalignment between the target and behavior policies using an overestimation/underestimation of value function at state $s^*$. Such $s^*$ is set as the best-case state associated with the highest value in our current estimation when computing upper bounds and the worst-case state estimate for lower bounds.

To formally introduce the estimation algorithm, we first introduce some necessary notations. Let $\boldsymbol{N}(s)$ denote the set of indices of episodes containing a state $s \in \mathcal{S}$, and let $\boldsymbol{t}_n(s)$ be the collection of time steps in the $n$-th episode such that for every $t \in \boldsymbol{t}_n(s)$, $s_t = s$. For any time step $t$, let $\pi_t = \pi(x_t \mid s_t)$ and $\neg\pi_t = 1 - \pi(x_t \mid s_t)$. We iteratively define the estimator for bounds over the state value function $V_\pi(s)$ as follows, for any state $s \in \mathcal{S}$,

$$\widehat{V_\pi}(s) = \frac{1}{N} \sum_{n \in \boldsymbol{N}(s)} \sum_{t \in \boldsymbol{t}(s)} \sum_{k=0}^{T_n - t} \gamma^k \Big(\pi_{t+k}y_{t+k} + \neg\pi_{t+k}\big(w + \gamma V(s^*)\big)\Big) \prod_{i=t}^{t+k-1} \pi_i, \tag{14}$$

Among the above equation, $N$ represents the total number of occurrences for the even $s_t = s$ in the observational data. we set parameters $w = a$ and $V(s^*) = \min_s V(s)$ when estimating the lower bound $\underline{V_\pi}(s)$; parameters $w = b$ and $V(s^*) = \max_s V(s)$ for the upper bound $\overline{V_\pi}(s)$.

An eligibility-trace version of our proposed estimation strategy is described Alg. 1. The algorithm keeps track of eligibility traces for every state in a similar manner to standard off-policy temporal difference algorithms. The main difference is that here the eligibility trace is multiplied by the target policy $\pi(x_{t-1} \mid s_{t-1})$ and a decay-rate $\lambda$, not including the nominal propensity score $P(x_{t-1} \mid s_{t-1})$. When computing the temporal difference error, the algorithm adjusts for the misalignment between the target and behavior policies by adding a regularized term $w + \gamma V_t(s^*)$, weighted by the probability $1 - \pi(x_t \mid s_t)$. We describe in Alg. 1 a version of C-TD($\lambda$) using *online update*. This means that the bounds estimate over value functions are updated at every time step. The *offline* version of the algorithm will use the same temporal difference error and eligibility traces. However, the update only occurs at the end of each episode; the increments and decrements are accumulated on the side, and the value function estimates do not change during the episode.

**Theorem 3.** *For any behavior policy, for any choice of $\lambda \in [0,1]$ that does not depend on the actions chosen at each state, let parameters $w$ and $s^*$ be defined as follows: (1) Lower Bound $\underline{V_\pi}$: $w = a$ and $s^* = \arg\min_s V_t(s)$; (2) Upper Bound $\overline{V_\pi}$: $w = b$ and $s^* = \arg\max_s V_t(s)$. Then, Alg. 1 with offline updating converges with probability $1$ to lower bound $\underline{V_\pi}$ and upper bound $\overline{V_\pi}$, respectively, under the usual step-size conditions on $\alpha$.*

The proof of Thm. 3 first shows a contraction property for estimates $\widehat{V}_\pi$, and then follows the general convergence theorem in (Jaakkola et al., 1994).

### 3.2 CAUSAL TREE BACKUP

The algorithm described so far focuses on the estimation of the state value functions. We next introduce a novel algorithm to bound the state-action value function $Q_\pi$ from finite observations.

Our algorithm is based on an augmentation on the standard tree backup (TB (Precup et al., 2000)), which we call the *causal tree backup* (C-TB($\lambda$)). The main idea of this new algorithm is illustrated in the backup diagram of Fig. 4. Similar to the standard tree backup, our algorithm updates the value estimates for the action selected by the behavior policy at each time step based on the subsequent reward and the current estimation for the value of the next state. The algorithm then forms a new estimate for the target value function, using the old value estimates for the actions not observed in the observational data and the new estimated value for $t$-he action taken by the behavior policy. On the other hand, the main differences include the following. (1) Eligibility traces will not only be weighted by the target policy $\pi(x_t \mid s_t)$ using the observed trajectories, but also an indicator function $\mathbb{1}_{x_{t-1}=x}$ returning 1 if the previous action $x_{t-1}$ coincides with the target action $x$. (2) When the behavior policy takes the same action $x_t = x$ as the target action, the update follows standard TB and uses the next sampled state $s_t$; when the sampled action $x_t \neq x$ differs from the target, our algorithm updates, instead, using the value function associated with the next worst-case or best-case state $s^*$, corresponding to the estimation of the lower bound and upper bound respectively. The $n$-step causal tree-backup estimator is defined as

Figure 4: Backup diagram for C-TB($\lambda$).

$$\widehat{Q}_\pi(s,x) = \frac{1}{N} \sum_{n \in \boldsymbol{N}(s)} \sum_{t \in \boldsymbol{t}(s)} \gamma^n Q(s_{t+n}, x_{t+n}) \prod_{i=t}^{t+n-1} \pi_{i+1} \mathbb{1}_{x_i=x} + \sum_{k=t}^{t+n} \gamma^{k-t+1} \prod_{i=t}^{t+k-1} \pi_{i+1} \mathbb{1}_{x_i=x}$$

$$\cdot \left( \mathbb{1}_{x_k=x} \left( y_k + \sum_{x' \neq x} \pi(x' \mid s_{k+1}) Q(s_{k+1}, x') \right) + \mathbb{1}_{x_k \neq x} \left( w + \sum_{x'} \pi(x' \mid s^*) Q(s^*, x') \right) \right) \quad (19)$$

The above tree backup estimator also has a simple incremental implementation using eligibility traces. An online version of this implementation is shown in Fig. 4.

**Theorem 4.** *For any behavior policy, for any choice of $\lambda \in [0,1]$ that does not depend on the actions chosen at each state, let parameters $w$ and $s^*$ be defined as follows: (1) Lower Bound $\underline{Q_\pi}$: $w = a$ and $s^* = \arg\min_s \sum_{x'} \pi(x' \mid s) Q_t(s, x')$; (2) Upper Bound $\overline{Q_\pi}$: $w = b$ and $s^* = \arg\max_s \sum_{x'} \pi(x' \mid s) Q_t(s, x')$. Then, Alg. 2 with offline updating converges with probability $1$ to lower bound $\underline{Q_\pi}$ and upper bound $\overline{Q_\pi}$, respectively, under the usual step-size conditions on $\alpha$.*

The proof of the above theorem relies on a contraction property on the estimates $\widehat{Q}_\pi$ and follows from the general convergence theorem in (Jaakkola et al., 1994).

## 4 EXPERIMENTS

We demonstrate our algorithms using different behavior policies in the Windy Gridworld described in Example 1. Overall, we found that simulation results support our findings, and the proposed algorithms consistently obtain informative bounds over value functions. Experiment 1 evaluates the performance of our bounding strategy in the presence of unobserved confounding. Experiment 2 uses data collected from a deterministic sub-optimal policy, violating the overlap. All experiments use

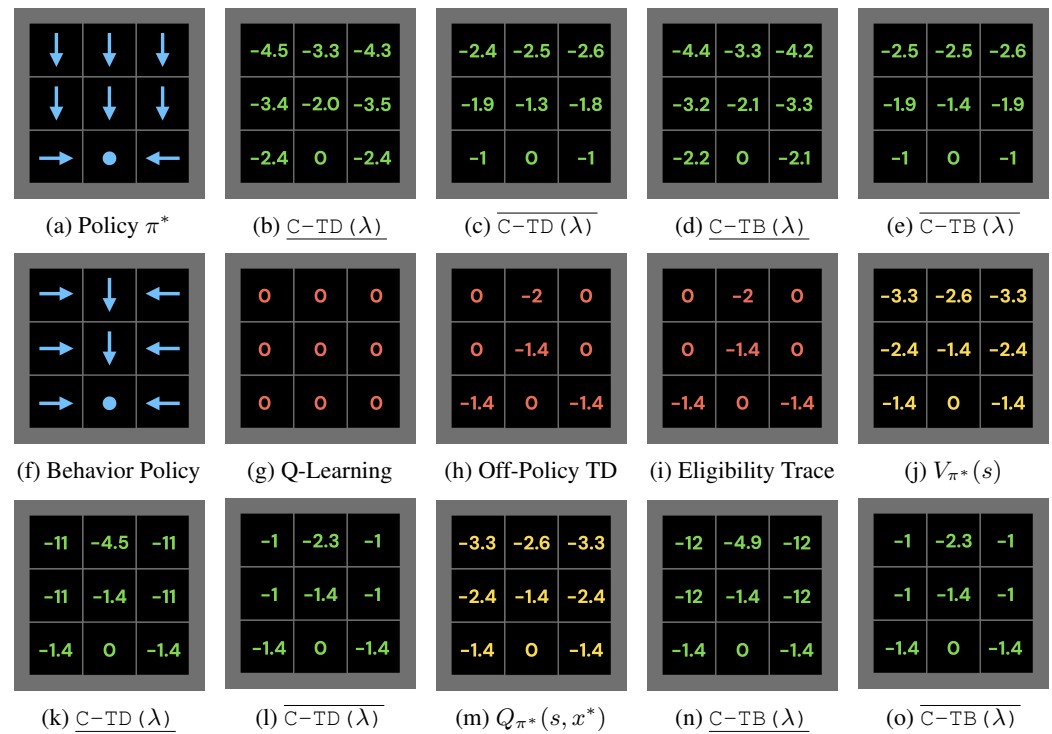

Figure 5: Simulation results comparing causally enhanced off-policy algorithms using eligibility traces ($\texttt{C-TD}(\lambda)$ and $\texttt{C-TB}(\lambda)$) with standard off-policy methods. The offline data are collected from (a - e) a confounded behavior policy affected by the unobserved confounder; and (f - o) a deterministic behavior policy following sub-optimal actions.

$5 \times 10^4$ offline observational samples, meaning that error bars are not significant, hence, not explicitly shown; the decay factor $\lambda = 0.5$. See Appendix B for more details on the experimental setup.

**Experiment 1.** Consider again the learning setting described in Example 1 where the offline data is contaminated with unobserved confounding bias, and the behavior policy selects actions based on the agent's state and the latent wind direction. We apply $\texttt{C-TD}(\lambda)$ to derive bounds over the state value function $V_{\pi^*}(s)$ and provide them in Figs. 5b and 5c. We also compute the bounds over the state-action value function $Q_{\pi^*}(s, x^*)$ for actions $x^* \leftarrow \pi^*(s)$ using $\texttt{C-TB}(\lambda)$; the simulation results are shown in Figs. 5d and 5e. The analysis reveals that our algorithm consistently recovers the closed-form bounds containing the ground-truth value functions, as previously shown in Fig. 2.

**Experiment 2.** For the Windy Gridworld environment described in Example 1, suppose the data is now collected by a deterministic behavior policy that always first moves towards the center and then moves down toward the goal; its parametrization is provided in Fig. 5f. This means that the overlap does not hold when the agent is located on either side of the top half of the board. We apply standard off-policy algorithms to evaluate the effect of the target policy $\pi^*$ of Fig. 5a and provide their evaluations in Figs. 5g to 5i. The propensity score is truncated using a small positive real $0 < \epsilon < 1$ if $P(x \mid s) = 0$. We also compute bounds over the target value functions using our proposed algorithms, $\texttt{C-TD}(\lambda)$ and $\texttt{C-TB}(\lambda)$, and provide their evaluations in Figs. 5k to 5l and Figs. 5n to 5o respectively. By comparing with the ground-truth values in Figs. 5j and 5m, we found that $\texttt{C-TD}(\lambda)$ and $\texttt{C-TB}(\lambda)$ can consistently obtain informative bounds; as expected, standard off-policy methods are not robust against no-overlap and deviate significantly from the target effects.

## 5 CONCLUSION

This paper investigates off-policy evaluation in Markov Decision Processes from offline data collected by a different behavior policy, where unobserved confounding bias and no-overlap cannot be ruled

out *a priori*. This leads to violations of causal consistency (Def. 2), which could pose significant challenges to standard off-policy algorithms. We first extend the celebrated Bellman's equation to derive informative bounds over values functions from the observational data, which are robust against bias due to the presence of unobserved confounding and no-overlap. Based on these extended equations, we propose two novel model-free off-policy algorithms using eligibility traces – one based on the standard temporal difference (`C-TD(λ)`), and the other based on the tree-backup (`C-TB(λ)`). These algorithms permit us to bound value functions from finite observations consistently.

## ETHICS STATEMENT

This paper investigates the theoretical framework of robust off-policy evaluation from biased offline data generated by a different behavior. Since unobserved confounding or no-overlap cannot be ruled out *a priori*, the agent's system dynamics in the environment cannot be fully identified from the offline data. To address this challenge, we proposed novel off-policy algorithms that allow the agent to derive informative bounds over value functions induced by a target policy from biased offline data. A positive impact of this work is that we address the potential risk of policy learning from offline data with the presence of unobserved confounding. Our framework is inherently robust against confounding bias and may apply to various consequential domains involving complex human interactions, including healthcare, marketing, finance, and autonomous driving. More broadly, automated decision systems using causal inference methods prioritize safety and robustness during their learning processes. Such requirements are increasingly essential since black-box AI systems are prevalent, and our understanding of their potential implications is still limited.

## REPRODUCIBILITY STATEMENT

The complete proof of all theoretical results presented in this paper, including Thms. 1 to 4, is provided in Appendix B. Detailed descriptions of the experimental setup are included in Appendix C. Readers can find all appendices as part of the supplementary text after the "References" section. All the experiments are synthetic and do not introduce any new assets. Windy Gridworld is implemented based on the Gymnasium framework (Towers et al., 2024).

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

# A    RELATED WORK

Our work builds upon the literature on the partial identification of causal effects, sensitivity analysis, and robust reinforcement learning from offline data.

**Partial Identification and Sensitivity Analysis**    Seminal work of Manski (1990) developed the first bounds on causal effects in non-identifiable settings using observational data in the single-stage treatment model with contextual information (i.e., a contextual bandit model). These bounds were then expanded to the instrumental variable setting (Balke & Pearl, 1997; Imbens & Angrist, 1994) partially identify counterfactual probabilities of causation (Tian & Pearl, 2000). More recently, (Zhang & Bareinboim, 2021) improved the bounds for applicability to continuous outcomes. (Zhang et al., 2022) established a general framework for estimating bounds on interventional and counterfactual effects. While Zhang et al. (2022) develop informative bounds using both observational and experimental data, they focus on general counterfactual queries by discretizing the exogenous latent space, formulating bounds as polynomial programs over this discretization and a Bayesian framework to approximately estimate bounds using MCMC.

Sensitivity analysis attempts to provide intervals on causal effects by assuming the level of confounding, for example, via models such as Marginal Sensitivity analysis, which considers deviations in the propensity score in relation to the estimated propensity (Rosenbaum, 2005; Richardson et al., 2014; Todem et al., 2010; Vansteelandt et al., 2006; Kallus & Zhou, 2018; Kallus et al., 2019; Namkoong et al., 2020; Jesson et al., 2022; Bruns-Smith & Zhou, 2023; Kausik et al., 2024). Other approaches explore additional parametric assumptions about the structural functions, including linearity (Cinelli et al., 2019) and Lipschitz continuity (Khan et al., 2023). Our work does not rely on additional functional constraints on the underlying system dynamics. Instead, we focus on the settings of standard discrete Markov Decision Processes (MDPs) with an infinite horizon. We develop robust off-policy evaluation algorithms to estimate closed-form bounds over the discounted cumulative rewards of candidate policies from offline observational data contaminated with unobserved confounding bias.

**Robust Reinforcement Learning**    Unlike planning in a standard MDP, robust reinforcement learning does not assume the parametrization of the transition probability function in the underlying model to be precisely determined. Instead, it is contained in a set of model parameters which is called the uncertainty set (Iyengar, 2005; Nilim & El Ghaoui, 2005; Xu & Mannor, 2010; Wiesemann et al., 2013; Yu & Xu, 2015; Mannor et al., 2016; Petrik & Russel, 2019). The goal of the agent is to learn a robust policy that performs the best under the worst possible case in the uncertainty set. Similar problems have been studied under the rubrics of safe policy learning (Thomas et al., 2015; Ghavamzadeh et al., 2016) or pessimistic reinforcement learning (Shi et al., 2022).[2]

Robust RL algorithms with provable guarantees have been proposed in tabular settings or under the assumptions of linear functions (Lim et al., 2013; Tamar et al., 2014; Roy et al., 2017; Badrinath & Kalathil, 2021; Wang & Zou, 2021). Combined with the computational framework of deep learning, robust RL algorithms have been extended to complex, high-dimensional domains (Pinto et al., 2017; Zhang et al., 2020). More recently, (Panaganti et al., 2022) proposed Robust Fitted Q-Iteration (RFQI) to learn the best possible robust policy from offline data with theoretical guarantees on the performance of the learned policy. Our work differs from robust RL methods since it does not require a pre-specified uncertainty set of model parameters. Instead, we construct the ignorance region over the underlying system dynamics from the confounded observational data using partial causal identification. Based on the learned uncertainty set, we then derived closed-form bounds over the value functions of the target policy. *To the best of our knowledge, this is the first work that develops off-policy algorithms using eligibility traces to obtain evaluations of candidate policies from biased offline data, possibly contaminated with unmeasured confounding or no-overlap, with provable guarantees on the convergence of learned evaluations.*

---

[2]Indeed, the idea of planning over a convex set of model parameters have been explored in online reinforcement learning. (Strehl & Littman, 2008) utilized an extended dynamic programming to learn an optimistic policy over a confidence set of models to balance the trade-off between exploration and exploitation.

## B   PROOFS

This section provides proof of the main theoretical results provided in the paper.

**Theorem 1** (Causal Bellman Equation). *For an MDP environment $M$ with reward $Y_t \in [a, b] \subseteq \mathbb{R}$, for any policy $\pi(x \mid s)$, its state value function $V_\pi(s) \in \left[\underline{V_\pi}(s), \overline{V_\pi}(s)\right]$ for every state $s \in \mathcal{S}$, where bounds $\underline{V_\pi}, \overline{V_\pi}$ are solutions given by the following dynamic programs,[3]*

$$\left\langle \underline{V_\pi}(s), \overline{V_\pi}(s) \right\rangle = \sum_x P(x \mid s) \left( \pi(x \mid s) \left( \widetilde{\mathcal{R}}(s, x) + \gamma \sum_{s', x'} \widetilde{\mathcal{T}}(s, x, s') \left\langle \underline{V_\pi}(s'), \overline{V_\pi}(s') \right\rangle \right) \right. \tag{10}$$

$$\left. + \pi(\neg x \mid s) \left( \langle a, b \rangle + \gamma \left\langle \min_{s'} \underline{V_\pi}(s'), \max_{s'} \overline{V_\pi}(s') \right\rangle \right) \right) \tag{11}$$

*Proof.* Following the Bellman equation (Bellman, 1966), the state value function at state $s \in \mathcal{S}$ is given by

$$V_\pi(s) = \sum_x \pi(x \mid s) \left( \mathcal{R}(s, x) + \gamma \sum_{s'} \mathcal{T}(s, x, s') V_\pi(s') \right) \tag{20}$$

Among the above quantities, the reward function $\mathcal{R}$ is bounded from the observational distribution (Manski, 1990) as follows,

$$\widetilde{\mathcal{R}}(s, x) P(x \mid s) + aP(\neg x \mid s) \leq \mathcal{R}(s, x) \leq \widetilde{\mathcal{R}}(s, x) P(x \mid s) + bP(\neg x \mid s) \tag{21}$$

where $\widetilde{\mathcal{R}}$ is the nominal reward function computed from the observational distribution and is defined in Eq. (9). Replacing the reward function $\mathcal{R}$ in Eq. (20) with the above lower bound gives

$$V_\pi(s) \geq \sum_x \pi(x \mid s) \left( \widetilde{\mathcal{R}}(s, x) P(x \mid s) + aP(\neg x \mid s) + \gamma \sum_{s'} \mathcal{T}(s, x, s') V_\pi(s') \right) \\ + \sum_x b\pi(x \mid s) P(\neg x \mid s) \tag{22}$$

Similarly, the transition distribution $\mathcal{T}$ can be bounded from the observational distribution (Manski, 1990),

$$\widetilde{\mathcal{T}}(s, x, s') P(x \mid s) \leq \mathcal{T}(s, x, s') \leq \widetilde{\mathcal{T}}(s, x, s') P(x \mid s) + P(\neg x \mid s) \tag{23}$$

and $\widetilde{\mathcal{T}}$ is the nominal transition distribution computed from the observational distribution defined in Eq. (9). Minimizing the lower bound in Eq. (22) subject to the above observational constraints in Eq. (23) and $\sum_{s'} \mathcal{T}(s, x, s') = 1$ gives the following lower bound:

$$V_\pi(s) \geq \sum_x \pi(x \mid s) P(x \mid s) \left( \widetilde{\mathcal{R}}(s, x) + aP(\neg x \mid s) + \gamma \sum_{s'} \widetilde{\mathcal{T}}(s, x, s') V_\pi(s') \right) \\ + \sum_x \pi(x \mid s) P(\neg x \mid s) \left( b + \min_{s'} V_\pi(s') \right) \tag{24}$$

The above lower bound is achieved by setting the worst-case transition probability $\mathcal{T}(s, x, s^*) = P(\neg x \mid s)$ for state $s^* = \arg\min_{s'} V_\pi(s')$ and $\mathcal{T}(s, x, s') = \widetilde{\mathcal{T}}(s, x, s') P(x \mid s)$ for all the other state $s' \neq s^*$. Note that the second term of the above inequality could be further written as:

$$\sum_x \pi(x \mid s) P(\neg x \mid s) \left( a + \min_{s'} V_\pi(s') \right) \tag{25}$$

$$= \sum_x \pi(x \mid s) \left( 1 - P(x \mid s) \right) \left( a + \min_{s'} V_\pi(s') \right) \tag{26}$$

$$= \sum_x \pi(x \mid s) \left( a + \min_{s'} V_\pi(s') \right) - \sum_x \pi(x \mid s) P(x \mid s) \left( a + \min_{s'} V_\pi(s') \right) \tag{27}$$

$$= \sum_x P(x \mid s) \left( a + \min_{s'} V_\pi(s') \right) - \sum_x \pi(x \mid s) P(x \mid s) \left( a + \min_{s'} V_\pi(s') \right) \tag{28}$$

---

[3] $\langle a, b \rangle$ is a vector containing a lower bound $a$ and an upper bound $b$. We highlight quantities that are different from the standard Bellmen Equation.

The last step holds since for any constant real value $C$, $\sum_x \pi(x \mid s)C = \sum_x P(x \mid s)C$. The above equation can be further written as

$$\sum_x \pi(x \mid s)P(\neg x \mid s)\left(a + \min_{s'} V_\pi(s')\right) = \sum_x \pi(\neg x \mid s)P(x \mid s)\left(a + \min_{s'} V_\pi(s')\right) \tag{29}$$

Replacing the second term in Eq. (24) gives

$$V_\pi(s) \geq \sum_x \pi(x \mid s)P(x \mid s)\left(\widetilde{\mathcal{R}}(s,x) + bP(\neg x \mid s) + \gamma \sum_{s'} \widetilde{\mathcal{T}}(s,x,s')V_\pi(s')\right) \\ + \sum_x \pi(\neg x \mid s)P(x \mid s)\left(a + \min_{s'} V_\pi(s')\right) \tag{30}$$

After a few simplifications, we obtain

$$V_\pi(s) \geq P(x \mid s)\left(\pi(x \mid s)\left(\widetilde{\mathcal{R}}(s,x) + \gamma \sum_{s',x'} \widetilde{\mathcal{T}}(s,x,s') V_\pi(s')\right) \right. \\ \left. + \pi(\neg x \mid s)\left(a + \gamma \min_{s'} V_\pi(s')\right)\right) \tag{31}$$

Finally, minimizing the value function $V_\pi$ subject to the above inequality gives the lower bound $\underline{V_\pi}$. The upper bound $\overline{V_\pi}$ over the state value function could be similarly derived. $\qquad\square$

**Theorem 2** (Causal Bellman Equation). *For an MDP environment $M$ with reward signals $Y_t \in [a,b] \subseteq \mathbb{R}$, for any policy $\pi(x \mid s)$, its state-action value function $Q_\pi \in \left[\underline{Q_\pi}(s,x), \overline{Q_\pi}(s,x)\right]$ for any state-action pair $(s,x) \in \mathcal{S} \times \mathcal{X}$, where bounds $\underline{Q_\pi}, \overline{Q_\pi}$ are given by as follows,*

$$\left\langle \underline{Q_\pi}(s,x), \overline{Q_\pi}(s,x)\right\rangle = P(x \mid s)\left(\widetilde{\mathcal{R}}(s,x) + \gamma \sum_{s',x'} \widetilde{\mathcal{T}}(s,x,s')\left\langle \underline{V_\pi}(s'), \overline{V_\pi}(s')\right\rangle\right) \tag{12}$$

$$+ P(\neg x \mid s)\left(\langle a,b\rangle + \gamma \left\langle \min_{s'} \underline{V_\pi}(s'), \max_{s'} \overline{V_\pi}(s')\right\rangle\right) \tag{13}$$

*Proof.* Applying Bellman equation (Bellman, 1966) allows us to iteratively write the state-action value function for any state-action pair $(s,x) \in \mathcal{S} \times \mathcal{X}$ as

$$Q_\pi(s,x) = \mathcal{R}(s,x) + \gamma \sum_{s'} \mathcal{T}(s,x,s')V_\pi(s') \tag{32}$$

where the reward function $\mathcal{R}$ is bounded from the observational distribution (Manski, 1990) following Eq. (21). Replacing the reward function $\mathcal{R}$ in the above equation with the corresponding lower bound gives

$$Q_\pi(s,x) \geq P(x \mid s)\left(\widetilde{\mathcal{R}}(s,x) + \gamma \sum_{s'} \mathcal{T}(s,x,s')V_\pi(s')\right) + aP(\neg x \mid s) \tag{33}$$

Similarly, the transition distribution $\mathcal{T}$ can be bounded from the observational distribution (Manski, 1990) following Eq. (23). Minimizing the lower bound in Eq. (33) subject to the above observational constraints in Eq. (23) and $\sum_{s'} \mathcal{T}(s,x,s') = 1$ gives the following solution:

$$Q_\pi(s,x) \geq P(x \mid s)\left(\widetilde{\mathcal{R}}(s,x) + \gamma \sum_{s'} \widetilde{\mathcal{T}}(s,x,s')V_\pi(s')\right) + P(\neg x \mid s)\left(a + \min_{s'} V_\pi(s')\right) \tag{34}$$

This lower bound is achieved by setting the worst-case transition probability $\mathcal{T}(s,x,s^*) = P(\neg x \mid s)$ for state $s^* = \arg\min_{s'} V_\pi(s')$ and $\mathcal{T}(s,x,s') = \widetilde{\mathcal{T}}(s,x,s')P(x \mid s)$ for all the other state $s' \neq s^*$. Finally, notice that $V_\pi(s)$ is a function of $Q_\pi(s,x)$ and is given by $V_\pi(s) = \sum_x \pi(x \mid s)Q_\pi(s,x)$. Minimizing the state-action value function $Q_\pi$ subject to the above inequality leads to the lower bound $\underline{Q_\pi}$. The upper bound $\overline{Q_\pi}$ could be similarly derived. $\qquad\square$

**Theorem 3.** *For any behavior policy, for any choice of $\lambda \in [0, 1]$ that does not depend on the actions chosen at each state, let parameters $w$ and $s^*$ be defined as follows: (1) Lower Bound $\underline{V_\pi}$: $w = a$ and $s^* = \arg\min_s V_t(s)$; (2) Upper Bound $\overline{V_\pi}$: $w = b$ and $s^* = \arg\max_s V_t(s)$. Then, Alg. 1 with offline updating converges with probability 1 to lower bound $\underline{V_\pi}$ and upper bound $\overline{V_\pi}$, respectively, under the usual step-size conditions on $\alpha$.*

*Proof.* We will focus on the convergence of lower bound $\underline{V_\pi}(s)$; the proof for the upper bound $\overline{V_\pi}(s)$ follows analogously. The proof is structured in two stages. First, we consider the truncated lower bound estimates corresponding to Eq. (14), which sums the adjusted rewards obtained from the environment for only $n$ steps, then uses the current estimate of the value function lower bound to approximate the remaining value:

$$\underline{R_t}^{(n)} = \sum_{k=0}^{n-1} \gamma^k \left( \pi_{t+k} y_{t+k} + \neg\pi_{t+k} \big( b + \gamma \min_{s'} V(s') \big) \right) \prod_{i=t}^{t+k-1} \pi_i + \gamma^n V(s_{t+n}) \prod_{i=t}^{t+k-1} \pi_i \quad (35)$$

We need to show that $\underline{R_t}^{(n)} - \underline{V_\pi}$ is a contraction mapping in the max norm. If this is true for any $n$, then by applying the general convergence theorem, the $n$-step return converges to $\underline{V_\pi}$. Then any convex combination will also converge to $\underline{V_\pi}$. For example, any combination using a $\lambda$ parameter in the style of eligibility traces will converge to $\underline{V_\pi}$.

The expected value of the adjusted return with regard to the observational distribution for state $s$ can be expressed as follows [4]:

$$\mathbb{E}\left[ \underline{R_t}^{(n)} \mid S_t = s \right] \quad (36)$$

$$= \sum_{k=1}^{n} \sum_{\bar{\boldsymbol{s}}_{1:k}, \bar{\boldsymbol{x}}_{1:k}, \bar{\boldsymbol{y}}_{1:k}} P\left( \bar{\boldsymbol{s}}_{1:k}, \bar{\boldsymbol{x}}_{1:k}, \bar{\boldsymbol{y}}_{1:k} \right) \gamma^{k-1} \left( \pi_k y_k + \neg\pi_k \left( b + \min_{s'} V(s') \right) \right) \prod_{i=1}^{k-1} \pi_i \quad (37)$$

$$+ \sum_{\bar{\boldsymbol{s}}_{1:n}, \bar{\boldsymbol{x}}_{1:n}} P\left( \bar{\boldsymbol{s}}_{1:n}, \bar{\boldsymbol{x}}_{1:n} \right) \gamma^n V(s_n) \prod_{i=1}^{n-1} \pi_i \quad (38)$$

$$= \sum_{k=1}^{n} \gamma^{k-1} \sum_{\bar{\boldsymbol{s}}_{1:k}, \bar{\boldsymbol{x}}_{1:k}} \prod_{i=1}^{k-1} \widetilde{T}(s_i, x_i, s_{i+1}) P(x_i \mid s_i) \pi(x_i \mid s_i) \quad (39)$$

$$\cdot \left( \pi(x_k \mid s_k) \widetilde{\mathcal{R}}(s_k, x_k) + \neg\pi(x_k \mid s_k) \left( b + \gamma \min_{s'} V(s') \right) \right) \quad (40)$$

$$+ \gamma^n \sum_{\bar{\boldsymbol{s}}_{1:n}, \bar{\boldsymbol{x}}_{1:n}} \prod_{i=1}^{n-1} \widetilde{T}(s_i, x_i, s_{i+1}) P(x_i \mid s_i) \pi(x_i \mid s_i) V(s_n) \quad (41)$$

By applying the extended Bellman equation for the lower bound $\underline{V_\pi}$ iteratively $n$ times, we obtain:

$$\underline{V_\pi}(s) = \sum_{k=1}^{n} \sum_{\bar{\boldsymbol{s}}_{1:k}, \bar{\boldsymbol{x}}_{1:k}} \gamma^{k-1} \prod_{i=1}^{k-1} \widetilde{T}(s_i, x_i, s_{i+1}) P(x_i \mid s_i) \pi(x_i \mid s_i) \quad (42)$$

$$\cdot \left( \pi(x_k \mid s_k) \widetilde{\mathcal{R}}(s_k, x_k) + \neg\pi(x_k \mid s_k) \left( b + \gamma \min_{s'} \underline{V_\pi}(s') \right) \right) \quad (43)$$

$$+ \gamma^n \sum_{\bar{\boldsymbol{s}}_{1:n}, \bar{\boldsymbol{x}}_{1:n}} \prod_{i=1}^{n-1} \widetilde{T}(s_i, x_i, s_{i+1}) P(x_i \mid s_i) \pi(x_i \mid s_i) \underline{V_\pi}(s_n) \quad (44)$$

Therefore,

$$\max_s \left| \mathbb{E}\left[ \underline{R_t}^{(n)} \mid S_t = s \right] - \underline{V_\pi}(s) \right| \leq \gamma \max_s \left| V(s) - \underline{V_\pi}(s) \right| \quad (45)$$

This means that any $n$-step return is a contraction in the max norm, and therefore, by applying (Jaakkola et al., 1994, Theorem 1), it converges to $\underline{V_\pi}(s)$.

---

[4]We abuse notation a bit and ignore the expected value operator $\mathbb{E}[\cdot]$ outside.

In the second stage, we show that by applying the updates of Alg. 1 for $n$ successive steps, we perform the same update by using the $n$-step adjusted return $\underline{R_t}^{(n)}$. The eligibility trace for state $s$ can be written as, for $t_n \in \boldsymbol{t}(s)$,

$$e_t(s) = \gamma^{t-t_n} \prod_{i=t_n+1}^{t} \pi_i. \tag{46}$$

We have

$$\sum_{k=1}^{n} e_{t+k-1}(s)\delta_{t+k-1}(s) \tag{47}$$

$$= \sum_{k=1}^{n} \gamma^{k-1} \prod_{i=t+1}^{t+k-1} \pi_i \Big( \pi_{t+k} \left( y_{t+k} + \gamma V(s_{t+k}) \right) + \neg\pi_{t+k} \left( b + \gamma \min_{s'} V(s') \right) \tag{48}$$

$$- V(s_{t+k-1}) \Big) \tag{49}$$

$$= \sum_{k=0}^{n-1} \gamma^k \Big( \pi_{t+k} y_{t+k} + \neg\pi_{t+k} \big( b + \gamma \min_{s'} V(s') \big) \Big) \prod_{i=t}^{t+k-1} \pi_i + \gamma^n V(s_{t+n}) \prod_{i=t}^{t+k-1} \pi_i \tag{50}$$

$$- V(s_t) \tag{51}$$

$$= \underline{R_t}^{(n)} - V(s_t) \tag{52}$$

Since $\texttt{C-TD}(\lambda)$ is equivalent to applying a convex mixture of $n$-step updates, and each update converges to correct lower bounds $\underline{V_\pi}$ for the state value functions, Alg. 1 converges to correct lower bounds as well. $\qquad\square$

**Theorem 4.** *For any behavior policy, for any choice of $\lambda \in [0,1]$ that does not depend on the actions chosen at each state, let parameters $w$ and $s^*$ be defined as follows: (1) Lower Bound $\underline{Q_\pi}$: $w = a$ and $s^* = \arg\min_s \sum_{x'} \pi(x' \mid s)Q_t(s, x')$; (2) Upper Bound $\overline{Q_\pi}$: $w = b$ and $s^* = \arg\max_s \sum_{x'} \pi(x' \mid s)Q_t(s, x')$. Then, Alg. 2 with offline updating converges with probability 1 to lower bound $\underline{Q_\pi}$ and upper bound $\overline{Q_\pi}$, respectively, under the usual step-size conditions on $\alpha$.*

*Proof.* We will focus on the convergence of lower bound $\underline{Q_\pi}(s, x)$; the proof for the upper bound $\overline{Q_\pi}(s, x)$ follows analogously. This proof is structured in two stages. Let $Q_n$ denote the $n$-step tree backup estimator defined in Eq. (19). First we show that $\mathbb{E}\left[Q_n(s,x)\right] - \underline{Q_\pi}(s,x)$ is a contraction using a proof by induction.

Let $Q$ be the current estimate of the lower bound for the value function. For $n = 1$,

$$\max_{s,x} \left| \mathbb{E}\left[ Q_1(s,x) \right] - \underline{Q_\pi}(s,x) \right| \tag{53}$$

$$= \max_{s,x} \left| P(x \mid s)\left( \widetilde{\mathcal{R}}(s,x) + \gamma \sum_{s',x'} \widetilde{\mathcal{T}}(s,x,s') \sum_{x'} \pi(x' \mid s')Q(s',x') \right) \right. \tag{54}$$

$$+ P(\neg x \mid s)\left( b + \gamma \min_{s'} \sum_{x'} \pi(x' \mid s')Q(s',x') \right) \tag{55}$$

$$- P(x \mid s)\left( \widetilde{\mathcal{R}}(s,x) + \gamma \sum_{s',x'} \widetilde{\mathcal{T}}(s,x,s') \sum_{x'} \pi(x' \mid s')\underline{Q_\pi}(s',x') \right) \tag{56}$$

$$\left. - P(\neg x \mid s)\left( b + \gamma \min_{s'} \sum_{x'} \pi(x' \mid s')\underline{Q_\pi}(s',x') \right) \right| \tag{57}$$

$$\leq \gamma \max_{s,x} \left| Q(s,x) - \underline{Q_\pi}(s,x) \right| \tag{58}$$

For the induction step, we assume that

$$\max_{s,x} \left| \mathbb{E}\left[ Q_n(s,x) \right] - \underline{Q_\pi}(s,x) \right| \leq \gamma \max_{s,x} \left| Q(s,x) - \underline{Q_\pi}(s,x) \right| \tag{59}$$

Next we want to show that the same holds for $Q_{n+1}(s, x)$. We can rewrite $Q_{n+1}(s, x)$ as follows,

$$Q_{n+1}(s, x) = \mathbb{1}_{x_t = x}\left( y_t + \sum_{x'}\left(\mathbb{1}_{x' \neq x}\pi(x' \mid s_{t+1})Q(s_{t+1}, x') + \mathbb{1}_{x' = x}Q_n(s_{t+1}, x)\right)\right) \quad (60)$$

$$+ \mathbb{1}_{x_t \neq x}\left(w + \sum_{x'}\pi(x' \mid s^*)Q(s^*, x')\right) \quad (61)$$

We must have

$$\max_{s,x}\left|\mathbb{E}\left[Q_{n+1}(s, x)\right] - \underline{Q}_\pi(s, x)\right| \quad (62)$$

$$= \max_{s,x}\left|P(x \mid s)\left(\widetilde{\mathcal{R}}(s, x) + \gamma\sum_{s',x'}\widetilde{\mathcal{T}}(s, x, s')\sum_{x'}\pi(x' \mid s')\right.\right. \quad (63)$$

$$\mathbb{1}_{x' \neq x}Q(s', x') + \mathbb{1}_{x' = x}\mathbb{E}\left[Q_n(s', x)\right]\Big) \quad (64)$$

$$+ P(\neg x \mid s)\left(b + \gamma\min_{s'}\sum_{x'}\pi(x' \mid s')Q(s', x')\right) \quad (65)$$

$$- P(x \mid s)\left(\widetilde{\mathcal{R}}(s, x) + \gamma\sum_{s',x'}\widetilde{\mathcal{T}}(s, x, s')\sum_{x'}\pi(x' \mid s')\underline{Q}_\pi(s', x')\right) \quad (66)$$

$$- P(\neg x \mid s)\left(b + \gamma\min_{s'}\sum_{x'}\pi(x' \mid s')\underline{Q}_\pi(s', x')\right)\Bigg| \quad (67)$$

$$\leq \gamma\max_{s,x}\left|P(x \mid s)\gamma\sum_{s',x'}\widetilde{\mathcal{T}}(s, x, s')\sum_{x'}\pi(x' \mid s')\mathbb{1}_{x' \neq x}\left(Q(s', x') - \underline{Q}_\pi(s', x')\right)\right. \quad (68)$$

$$+ \mathbb{1}_{x' = x}\mathbb{E}\left[\left(Q_n(s', x) - \underline{Q}_\pi(s', x')\right)\right] \quad (69)$$

$$+ P(\neg x \mid s)\min_{s'}\sum_{x'}\pi(x' \mid s')\left(Q(s', x') - \underline{Q}_\pi(s', x')\right)\Bigg| \quad (70)$$

$$\leq \gamma\max_{s,x}\left|Q(s, x) - \underline{Q}_\pi(s, x)\right| \quad (71)$$

By applying (Jaakkola et al., 1994, Theorem 1), we can conclude that any $n$-step adjusted return converges to the correct lower bound for the state-action value function. Since all the n-step returns converge to $\underline{Q}_\pi$, any convex linear combination of $n$-step returns also converges to $\underline{Q}_\pi$.

For the second part of the proof, we show that `C-TB(`$\lambda$`)` with $\lambda = 1$ for n steps is equivalent to using $Q_n$. The eligibility trace for a state-action pair $(s, x)$ can be rewritten as:

$$e_t(s, x) = \gamma^k\prod_{i=t+1}^{t+k-1}\pi_{i+1}\mathbb{1}_{x_i = x}. \quad (72)$$

By adding and subtracting the weighted action value $\pi_{t+k}\mathbb{1}_{x_{t+k} = x}$ for the action taken on each step from the return, and regrouping, we have

$$Q(s_t, x) + \sum_{k=1}^{n}\gamma^{k-1}\prod_{i=t+1}^{t+k-1}\pi_{i+1}\mathbb{1}_{x_i = x}\left(\mathbb{1}_{x_{t+k} = x}\left(y_{t+k} + \sum_{x' \neq x}\pi(x' \mid s_{t+k+1})Q(s_{t+k+1}, x')\right)\right.$$

$$\quad (73)$$

$$+ \mathbb{1}_{x_{t+k} \neq x}\left(w + \min_{s'}\sum_{x'}\pi(x' \mid s')Q(s', x')\right) - Q(s_{t+k}, x)\Bigg) \quad (74)$$

$$= Q(s_t, x) + \sum_{k=1}^{n}e_{t+k}(s_t, x)\delta_{t+k}(x) \quad (75)$$

This concludes the proof. □

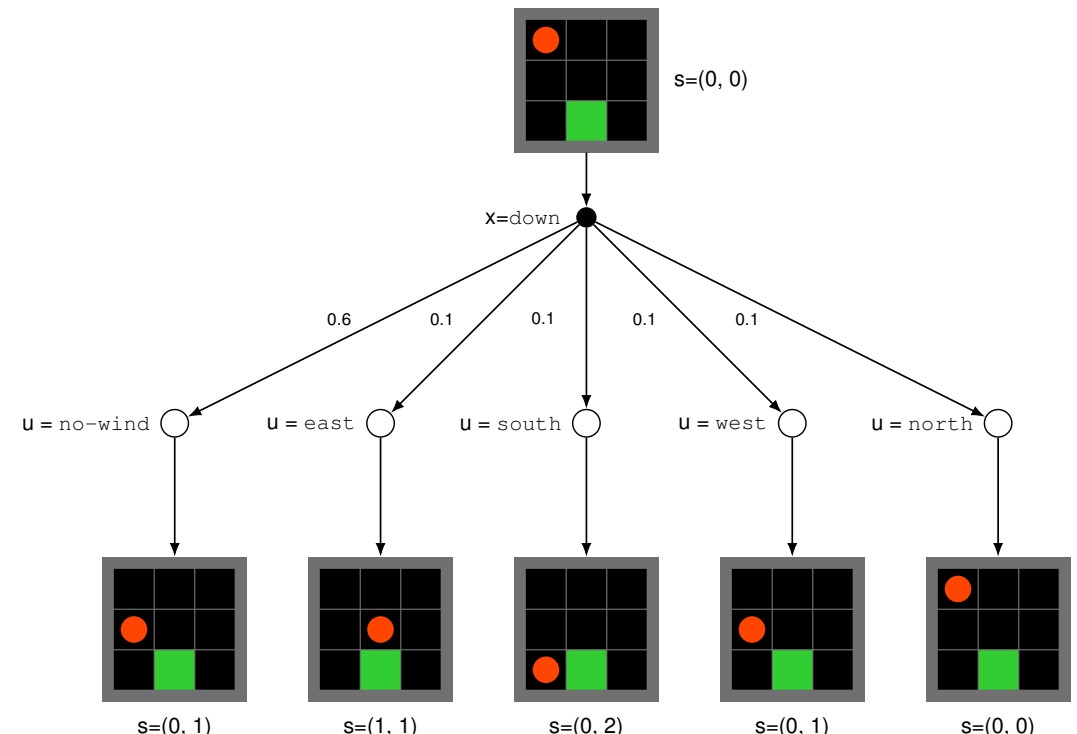

Figure 6: Trajectories sampled from the interventional transition distribution $\mathcal{T}$.

## C EXPERIMENTAL SETUPS

In this section, we provide details on the experimental setups and additional discussion on the simulation environment. All experiments were performed on a 2021 MacBook Pro with 16GB memory, implemented in Python. The simulation environment is built upon the Gymnasium framework (Brockman et al., 2016). We plan to release the source code with the camera-ready version of the manuscript.

**Windy Gridworld** Our simulation builds on the Windy Gridworld environment described in Fig. 1b, where the red dot represents the agent and the green square represents the goal state. The agent's location is represented using a vector $(i, j)$ where $i \in \{0, 1, 2\}$ is the column index, and $j \in \{0, 1, 2\}$ is the row index. So the agent's starting state is $(0, 0)$ and the goal state is $(1, 2)$. Fig. 7 shows the detailed state representation for each location in the gridworld.

The agent can take five actions $x \in \mathcal{X}$ - up, down, right, left, and stay-put, corresponding to vector $(0, -1)$, $(0, 1)$, $(1, 0)$, $(-1, 0)$, and $(0, 0)$ respectively. Meanwhile, the agent's movement is also affected by a wind; the wind direction $u \in \mathcal{U}$ include - north, south, east, west, and no-wind, corresponding to vector $(0, -1)$, $(0, 1)$, $(1, 0)$, $(-1, 0)$, and $(0, 0)$ respectively. Table 1 summarizes the detailed parametrization for the agent's action and the wind direction. For

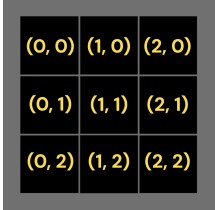

Figure 7: Agent's state in Windy Gridworld environment.

| Action $x$ | up | down | right | left | stay-put |
|---|---|---|---|---|---|
| Wind $u$ | north | south | east | west | no-wind |
| Vector $v$ | $(0, -1)$ | $(0, 1)$ | $(1, 0)$ | $(-1, 0)$ | $(0, 0)$ |

Table 1: Vector representations for the agent's action $X$ and the wind direction $U$.

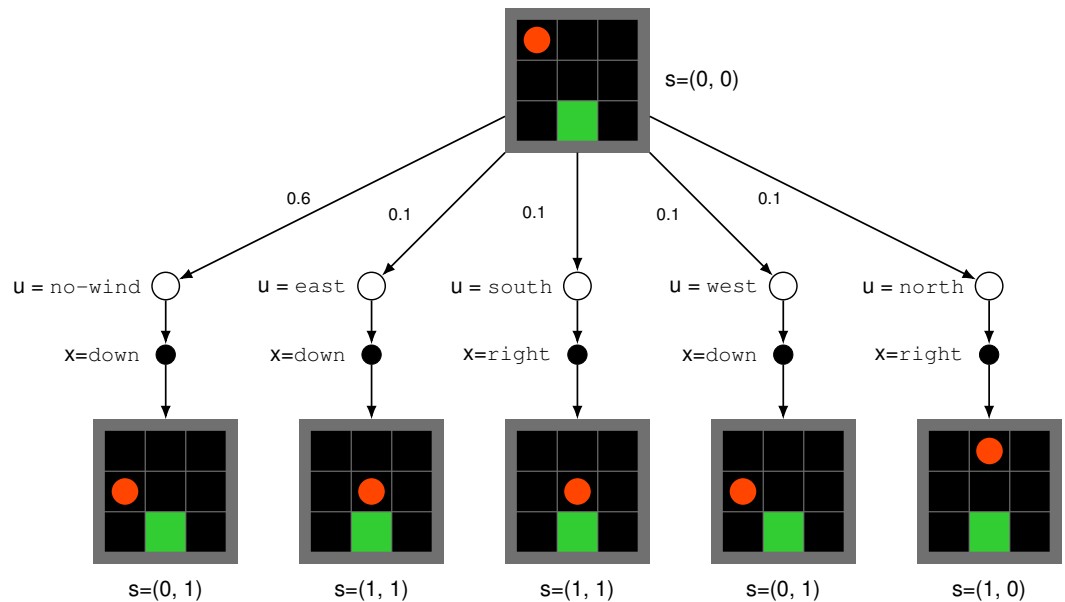

Figure 8: Trajectories sampled from the observational transition distribution $\widetilde{\mathcal{T}}$ induced by a con-founded behavior policy $f_X$.

every time step $t = 1, 2, \ldots$, the wind $U_t$ can blow in directions north, south, east, west with equal probabilities of $10\%$; otherwise, the weather is nice and there is no-wind. That is,

$$\forall i \in \{-1, 1\}, \quad P\left(U_t = (i, 0)\right) = P\left(U_t = (0, i)\right) = 0.1, \quad \text{and} \quad P\left(U_t = (0, 0)\right) = 0.6 \quad (76)$$

At every time step $t$, the agent receives a constant reward $Y_t \leftarrow -1$. The next state of the agent is shifted by both its action and the wind direction through the mechanism

$$S_{t+1} \leftarrow \max\left\{\min\left\{S_t + X_t + U_t, (2, 2)\right\}, (0, 0)\right\}. \quad (77)$$

In other words, the agent's next state $S_{t+1}$ is a vector sum of the agent's current location $S_t$, its action $X_t$, and the wind direction $U_t$, truncated by the board's boundary $i = 0, 2$ and $j = 0, 2$. For instance, we show in Fig. 6 the system dynamics for the agent's interactions with the gridworld environment at from the location $s = (0, 0)$, taking the action down ($x = (0, 1)$). In this case, when the wind is blowing towards south ($u = (0, 1)$), the agent's location will be shifted by both the action $x$ and the windy direction $u$, and moves to the bottom left corner $s' = (0, 2)$ at the next time step. Since among all wind directions, $u = $ east is the only latent state moving the agent to the center $s' = (0, 2)$, we must have the following evaluation for the interventional distribution $P_{X_t}\left(S_{t+1} \mid S_t\right)$,

$$P_{X_t \leftarrow (0,1)}\left(S_{t+1} = (0, 2) \mid S_t = (0, 0)\right) = P\left(U_t = (1, 0)\right) \quad (78)$$
$$= 0.1 \quad (79)$$

That is, the agent's transition distribution $\mathcal{T}(s, x, s') = 0.1$ when starting from $s = (0, 1)$, taking action $x = (0, 1)$, and moving to the next state $s' = (0, 2)$.

**Confounded Behavior Policy**  Consider now an off-policy learning task in the windy gridworld, where the agent's goal is to evaluate the expected return of a target policy $\pi^*$ described in Fig. 2a. Following such a policy $\pi^*$, the agent will consistently move towards the goal state $s = (1, 2)$ from its current location, regardless of the wind direction.

The detailed parametrization of the agent's system dynamics in the windy gridworld remains unknown. Instead, its has access to observed trajectories generated by a behavior policy $x \leftarrow f_X(s, u)$ which could sense the wind and select an action accordingly; Fig. 9 provides a detailed description for this behavior policy. For example, when the agent is located in the top-left corner ($s = (0, 0)$) and the wind is blowing south ($s = (0, 1)$), the behavior policy $x \leftarrow f_X(s, u)$ will decide to move right ($x = (1, 0)$) so that the agent could get to the center ($s' = (1, 1)$).



(a) `no wind`  (b) `east`  (c) `south`  (d) `west`  (e) `north`

Figure 9: A confounded behavior policy $f_X$ selecting values based on the agent's location $S$ and the latent wind direction $U$.

Consequently, the wind direction $U_t$ becomes an unobserved confounder in the generative process for the offline observational data, affecting the allocated action $X_t$ and the next state $S_{t+1}$ simultaneously. The presence of unobserved confounders lead to violations of causal consistency (Def. 2). To witness, Fig. 8 shows observed trajectories in the offline data when the agent starts from state $s = (0, 0)$. When the weather is nice (`no-wind`) or the wind $u$ is blowing towards `east` or `west`, the behavior policy selects action $x = $ `down`, similar to the interventional trajectories of Fig. 6. On the other hand, when the wind is blowing towards `north` or `south`, the behavior policy selects action $x = $ `right`, moving the agent towards the center of the board. Among all the possible next state in the observational data, we find that the agent will never reach the bottom left corner $s = (0, 2)$. This means that when evaluating the observational distribution $P(S_{t+1} \mid S_t, X_t)$, we must have

$$P(S_{t+1} = (0, 2) \mid S_t = (0, 0), X_t = (0, 1)) = 0 \qquad (80)$$

In other words, the nominal transition distribution $\widetilde{\mathcal{T}}(s, x, s') = 0$ when one observes the agent starting from $s = (0, 1)$, taking action $x = (0, 1)$, and moving to the next state $s' = (0, 2)$. Comparing the evaluations in Eqs. (79) and (80), we find that $P_{x_t}(s_{t+1} \mid s_t) \neq P(s_{t+1} \mid s_t, x_t)$, that is, causal consistency (Def. 2) does not hold between the agent's system dynamics in windy gridworld and the observational distribution generated by the confounded behavior policy in Fig. 9.

