# OpenReview forum: "Eligibility Traces for Confounding Robust Off-Policy Evaluation: A Causal Approach"
_ICLR.cc/2025/Conference — Submitted to ICLR 2025_

### Official Review · Reviewer_Srdk · 2024-10-29

**Soundness:** 2
**Presentation:** 3
**Contribution:** 2
**Rating:** 3
**Confidence:** 4

**Summary:**

This paper studies off-policy evaluation with offline data in Markov Decision Processes, where the actions taken by the behavior policy may be affected by unobserved confounders, causing standard estimation techniques to fail. The authors propose a variant of the Bellman equation that takes this confounding into account, and show they are able to obtain a consistent estimate of it from data. They demonstrate their approach on a gridworld experiment.

**Strengths:**

There is limited work on causality and handling unobserved confounders in RL, but this is an important problem, with possible practical applications. This work takes a step towards deepening our understanding of how we should think about RL in these settings.

**Weaknesses:**

1. The theoretical results, while a good first step, need further refinement to be convincing. In particular, the following aspects could be improved:
	* Theorem 1 and 2 give upper and lower bounds on the value and Q-value function in the confounded setting, but it is unclear how tight these bounds are. Can we obtain tighter upper and lower bounds, or is this the tightest possible? Is it possible to come up with a clean bound on the gap between the upper and lower bounds? Without answers to these questions, it is difficult to see how significant Theorem 1 and 2 are.
	* Theorem 3 and 4 are asymptotic consistency results. What are the finite-time properties of Algorithm 1? That is, for a fixed number of samples $n$, how small is the estimation error on the value function? While an asymptotic consistency result is nice, more refined analysis of this is required in order to show how practical this approach is.

2. The experimental results are limited to an extremely simple 3x3 grid world environment. Given the aforementioned shortcomings of the theoretical results, these experiments are not sufficient for illustrating the effectiveness of the proposed approach. More extensive experiments on more complex environments are necessary here with the current theoretical results.

3. Several notational issues. In particular, the $\langle$, $\rangle$ notation in Theorem 1 and 2 is not defined. I believe this is attempting to simultaneously state the upper and lower bounds, but unless I missed it, this was not stated. This should be clarified. It was also unclear and somewhat distracting why in Theorem 1 and Theorem 2 some of the font is blue.

4. More practical justification for why this problem is important would help better motivate the paper.

5. There are a variety of existing works on causality in bandits and RL that are not mentioned or cited here. See works [1]-[5] given below. These should be cited, and some discussion given of relation to the current work.

[1] Lattimore, Finnian, Tor Lattimore, and Mark D. Reid. "Causal bandits: Learning good interventions via causal inference." Advances in neural information processing systems 29 (2016).

[2] Lee, Sanghack, and Elias Bareinboim. "Structural causal bandits: Where to intervene?." Advances in neural information processing systems 31 (2018).

[3] Lu, Yangyi, Amirhossein Meisami, and Ambuj Tewari. "Causal bandits with unknown graph structure." Advances in Neural Information Processing Systems 34 (2021): 24817-24828.

[4] Lu, Chaochao, Bernhard Schölkopf, and José Miguel Hernández-Lobato. "Deconfounding reinforcement learning in observational settings." arXiv preprint arXiv:1812.10576 (2018).

[5] Wang, Lingxiao, Zhuoran Yang, and Zhaoran Wang. "Provably efficient causal reinforcement learning with confounded observational data." Advances in Neural Information Processing Systems 34 (2021): 21164-21175.

**Questions:**

See weaknesses section.

---

> ### Author Response · Authors · 2024-11-21
> **Response to Reviewer Srdk [1/2]**
>
> We appreciate the reviewer’s feedback. We believe that a few misunderstandings of our work led to some of the evaluations being overly harsh and would sincerely ask the reviewer to reconsider our paper given the clarifications provided below.
>
> ---
> >**W1(a)** _“Theorem 1 and 2 give upper and lower bounds on the value and Q-value function in the confounded setting, but it is unclear how tight these bounds are. Can we obtain tighter upper and lower bounds, or is this the tightest possible? Is it possible to come up with a clean bound on the gap between the upper and lower bounds? Without answers to these questions, it is difficult to see how significant Theorem 1 and 2 are.”_
>
> Regarding the tightness of Theorem 1 and 2, as stated in Lines 284-286, “Since Thms. 1 and 2 are closed-form solutions of optimization programs and the observational constraints in Eqs. (7) and (8) are tight, the extended Bellman’s equation bounds are sharp from offline data and Markov property. This means they cannot be improved without additional assumptions.”
>
> Could the reviewer also clarify what “clean bound” means? Theorems 1 and 2 provide closed-form bounds over the value functions associated with the target policy. It is defined recursively since we want to evaluate the discounted cumulative reward over an infinite horizon. This is similar to the standard value iteration and Q learning in MDP for unbiased offline data.
>
> ---
> >**W1(b)** _“Theorem 3 and 4 are asymptotic consistency results. What are the finite-time properties of Algorithm 1? That is, for a fixed number of samples , how small is the estimation error on the value function? While an asymptotic consistency result is nice, more refined analysis of this is required in order to show how practical this approach is.”_
>
> Our convergence analysis follows the framework of (Jaakkola et al., 1994), which was used to prove the convergence of the standard eligibility trace algorithms $TD(\lambda)$ in (Precup et al., 2000). As far as we are aware, Theorems 3 and 4 prove, for the first time, the convergence of model-free off-policy algorithms with eligibility traces that can sufficiently bound the target value functions from confounded offline data. We are interested in exploring the finite-time analysis of our proposed algorithms. It would be appreciated if the reviewer could provide references for the finite-time analysis for the convergence of the standard eligibility trace algorithms, e.g., (Precup et al., 2000).
>
> ---
> >**W2** _The experimental results are limited to an extremely simple 3x3 grid world environment. Given the aforementioned shortcomings of the theoretical results, these experiments are not sufficient for illustrating the effectiveness of the proposed approach. More extensive experiments on more complex environments are necessary here with the current theoretical results.”_
>
> This paper expands standard off-policy evaluation algorithms to situations where Causal Consistency (Def. 2) does not hold, and unobserved confounding bias is typically present in the offline data. To our knowledge, popular reinforcement learning benchmarks do not explicitly address the challenges posed by unobserved confounding. For this reason, we developed a synthetic evaluation environment in Windy Gridworld. We recognize that creating more complex evaluation benchmarks that accurately reflect the challenges of data bias is an exciting challenge and is under active development.
>
> ---
> >**W3** _“Several notational issues. In particular, the , notation in Theorem 1 and 2 is not defined. I believe this is attempting to simultaneously state the upper and lower bounds, but unless I missed it, this was not stated. This should be clarified. It was also unclear and somewhat distracting why in Theorem 1 and Theorem 2 some of the font is blue.”_
>
> $\langle \underline{V}, \overline{V} \rangle$ is an vector containing a pair representing the lower bound $ \underline{V}$ and upper bound $\overline{V}$. Theorems 1 and 2 generalize the standard Bellman equation to bound value function in non-identifiable settings. The highlighted part indicates the difference compared to the standard Bellman equation. We have updated the manuscript to provide additional explanations.

---

> ### Author Response · Authors · 2024-11-21
> **Response to Reviewer Srdk [2/2]**
>
> ---
> >**W4** _“More practical justification for why this problem is important would help better motivate the paper.”_
>
> The challenges of unobserved confounding and no-overlap have been widely acknowledged in healthcare, social science, econometrics, biostatistics, etc. For instance, when analyzing electronic healthcare records (EHR), physicians often prescribe treatments based on the patient’s socioeconomic status, which was not recorded in the data. This leads to unobserved confounding bias in EHRs, making the treatment appear to be more effective. More recently, there has been a growing body of work investigating robust reinforcement learning in the face of unobserved confounding and no-overlap, e.g.,  Kallus and Zhou (2018), Zhang and Bareinboim (2019), and Namkoong (2020). This paper proposes the first mode-free algorithms to reliably bound the target value functions from biased, offline data.
>
> ---
> >**W5** _“There are a variety of existing works on causality in bandits and RL that are not mentioned or cited here. See works [1]-[5] given below. These should be cited, and some discussion given of relation to the current work.”_
>
> We thank the reviewers for the references and are familiar with them. However, these works are orthogonal to our learning setting here. More specifically, [1-3] studies online reinforcement learning in bandit models; [4-5] studies off-policy learning in identifiable settings where the target expected reward is uniquely discernible from the data and causal assumptions. In this paper, we study off-policy evaluation on MDPs with an infinite horizon where the target effect is not identifiable from confounded data. We include a comprehensive discussion on related work in causal inference and reinforcement learning in Appendix A. We welcome the reviewer to check the appendices to see if our statements are true.

---

> > ### Author Response · Authors · 2024-11-27
> >
> > Thank you once again for your valuable feedback. We hope that our response has addressed your concerns, especially regarding the notations and related work in causal reinforcement learning. As we haven’t heard back from you, we would like to double-check if you have any additional comments or questions. We would be happy to address any further inquiries you have.

---

> > > ### Comment · Reviewer_Srdk · 2024-11-27
> > > **Response**
> > >
> > > Thanks to the authors for their reply. To clarify what I mean by a "clean bound on the gap between the upper and lower bounds", I am referring to whether we could obtain a bound on $\bar{V}\_\pi(s) - \underline{V}\_\pi(s)$ more interpretable than what is given in Theorem 1---for example, rather than just stating a recurrence, if we could bound this difference in terms of some global measure of causal inconsistency, or other standard quantities common in the RL literature such as the state-action value gaps.
> > >
> > > I do not have any further questions at this point. I am still concerned, however, that the experiments are overly simplistic and the theoretical results lack significance. I would like to maintain my original score.

---

> ### Author Response · Authors · 2024-11-28
> **Some misunderstandings [1/2]**
>
> We thank the reviewer for the engagement. However, there seem to be some misunderstandings in the basic problem setting and technical contribution.
>
> ---
> >**1.** _"if we could bound this difference in terms of some global measure of causal inconsistency, or other standard quantities common in the RL literature such as the state-action value gaps.”_
>
> First, there might be some confusion between the bounds in Theorems 1-2 and the regret bound in online learning. Our bounding results (Theorems 1 and 2) are off-policy algorithms that allow the agent to evaluate candidate policies from offline observations collected by other agents or behavioral policies. This differs from the regret-bound analysis for online RL algorithms (e.g., UCRL or UCB-VI) where the agent repeatedly performs direct actions in the underlying environment and adjusts its exploration policies based on observed outcomes. In this case, a typical instance-dependant regret bound accounts for the detailed parametrization of the underlying environment, e.g., the state-action value gaps measuring the distance between optimal and sub-optimal actions in the **ground-truth value functions**.
>
> On the other hand, Theorems 1 and 2 provide causal-effect bounds [1] that permit the agent to evaluate the ground-truth value functions from biased offline data. In the standard off-policy RL setting, the transition probabilities $\mathcal{T}(s, x, s’)$ and reward function $\mathcal{R}(s, x)$ can be fully determined from observed data by importance sampling or adjustment. The target value functions could be computed based on the iterative process in Bellman’s equation.
>
> 1: The terminology we used follows from the causal inference literature, including Manski (1989), Balke and Pearl (1997).
>
> In practice, however, when unobserved confounders exist and Causal Consistency (Def. 2) does not hold, the system dynamics $\mathcal{T}(s, x, s’), \mathcal{R}(s, x)$ are generally under-determined from the observed data. Consequently, the ground-truth value functions are also not identifiable from the observed data, which is the departing point of our work. This observation is part of the beauty of this work, building on the knowledge that observational and interventional distributions are different from the causal inference literature. To address this challenge of non-identifiability, we proposed novel off-policy algorithms in Theorems 1 and 2 to bound target value functions from observed data (since point identification is not achievable). These algorithms are iterative, generalizing the well-celebrated Bellman’s equation to biased, confounded data.
>
> More importantly, the bounding results in Theorems 1 and 2 are **not** regret bound and should not depend on the detailed parametrization of the underlying instance, e.g.,  the state-action value gaps. Since we focus on off-policy evaluation, our bounds in Theorems 1 and 2 only depend on the observational data. This permits us to design empirical procedures to estimate these bounds from finite samples. Taking one step back and acknowledging the importance of Bellman’s equation, which applies when unobserved confounding is not present, we believe our results constitute a major contribution to the literature, moving the use of RL (and Bellman equations) towards more realistic, practical settings in which confounding bias cannot be ruled out a prior.

---

> ### Author Response · Authors · 2024-11-28
> **Some misunderstandings [2/2]**
>
> ---
> >**2.** _"I am referring to whether we could obtain a bound on $\overline{V_{\pi}}(s) - \underline{V_{\pi}}(s)$ more interpretable than what is given in Theorem 1.”_
>
> As mentioned in our previous response, we intentionally designed the bounds in Theorem 1 to be recursive since we want to generalize the well-celebrated Ballman’s equation. A good mental image to interpret these bounds is the backup diagram in Fig. 3. As stated in Lines 320-355, “Similar to the standard off-policy _TD_, our algorithm will update the estimation of state value functions $\underline{V_{\pi}}, \overline{V_{\pi}}$ using the sampled trajectories of transitions in the observational data. ... Different from the standard off-policy _TD_, our proposed algorithm does not weight each step of the transition using importance sampling (or equivalently, inverse propensity weighting) since the true behavior policy $f_X$ (propensity score) is not recoverable from the observational data. Instead, _C-TD_ weights each transition using the target policy $\pi$ and adjusts for the misalignment between the target and behavior policies using an overestimation/underestimation of value function at state $s^*$. Such $s^*$ is set as the best-case state associated with the highest value in our current estimation when computing upper bounds and the worst-case state estimate for lower bounds.”
>
> ---
> >**3.** _"I am still concerned, however, that the experiments are overly simplistic and the theoretical results lack significance.”_
>
> We would like to clarify the significance of our foundational results further. Standard off-policy evaluation algorithms (e.g., TD) rely on the critical assumption of Causal Consistency, which excludes unobserved confounding bias and non-overlap. However, this assumption does not generally hold in many practical domains, e.g., healthcare and marketing, where one does not directly control the action assignment and the behavioral policy is not fully recoverable from data. To address these challenges, we propose off-policy evaluation methods in Theorems 1 and 2 which bound the value functions of candidate policies from biased, observational distribution, extending the well-celebrated Bellman’s equation. We then develop model-free temporal difference algorithms (Algs. 1 and 2) using eligibility traces to estimate these value function bounds from finite observations. Finally, we also provide theoretical analysis in Theorems 3 and 4, showing that our proposed algorithms are guaranteed to converge to the ground-truth value functions. As far as we are aware, these results are novel and extend the standard off-policy learning, for the first time, to biased data contaminated with unobserved confounding and no-overlap. Considering the significance of off-policy evaluation in RL literature and the prevalence of unobserved confounding bias in practice, we respectfully ask the reviewer to re-evaluate the significance of our work.

---

> > ### Author Response · Authors · 2024-12-02
> > **Summary of Response**
> >
> > Thank you once again for your engagement with our work. We hope that we have addressed your questions regarding the significance of our technical contributions. As we have yet to receive a response from you and the discussion period is nearing its end, we would like to briefly reiterate some critical points from our submissions, summarized below.
> >
> >  - _Clean bound_: This paper focuses on the off-policy evaluation setting, where the goal is to evaluate candidate policies using fixed, offline data. Consequently, our bounds over the target value function (Theorems 1 and 2) only take the observed data as input rather than rely on the detailed parametrization of the underlying instance, e.g., the state-action value gaps.
> >  - _Significance_: This paper proposes model-free off-policy evaluation algorithms, using eligibility traces, that are robust against biased offline data, due to the unobserved confounding and no-overlap. As far as we are aware, these algorithms are the first of their kind. Taking one step back and acknowledging the importance of off-policy learning and temporal difference algorithms (e.g., Q-Learning), which applies when unobserved confounding is not present, we believe our results constitute a significant contribution to the literature, moving the use of RL toward more realistic, practical settings in which confounding bias cannot be ruled out a priori.
> >
> > We sincerely hope you will reconsider our work in light of the above clarifications. We are also happy to address any further inquiries you may have.
> >
> > Authors of Paper #4929

---

### Official Review · Reviewer_nUhY · 2024-11-03

**Soundness:** 3
**Presentation:** 2
**Contribution:** 3
**Rating:** 6
**Confidence:** 3

**Summary:**

This paper addresses the challenges of off-policy evaluation in reinforcement learning (RL) when faced with confounded offline data and non-overlapping support between behavior and target policies. In such cases, traditional methods struggle to produce accurate value estimates due to unobserved confounders and the lack of common support, resulting in biased evaluations. The authors propose a novel model-free approach leveraging eligibility traces for partial identification of policy values that gives upper and lower bounds of the underlying true expected returns.

**Strengths:**

1. The problem of confounding in OPE studied by this paper is well motivated and is an important topic towards reliable and robust RL.
2. The model-free approach by leveraging temporal difference learning and eligibility traces for partial identification in OPE is new and interesting.
3. Theoretical results prove the convergence of the proposed algorithms to the partial identification interval given exact observational distributions.

**Weaknesses:**

1. The experiments of the proposed methods are limited to simple synthesis setups.
2. Lacking empirical comparisons with extensive body of partial identification OPE methods as mentioned in the related work section.

**Questions:**

1. How does the proposed algorithms perform on large-scale RL experiments? A direct difficulty of scaling up the algorithm is the need of solving $\min_{s\in\mathcal{S}}V(s)$ and $\max_{s\in\mathcal{S}}V(s)$ for some value estimate $V$.
2. Despite that the true partial identification interval (defined through Theorem 1) gives valid upper and lower bounds of the true policy value, it seems that the resulting bound could be too optimistic or too pessimistic since the observational data can be induced by arbitrarily bad behavior policy (and sure this makes sense). Is it possible to provide further analysis how does the behavior policy influence the accuracy of the true partial identification interval? Or are there methods to avoid such kind of potential looseness under some circumstances?
3. In addition to the asymptotic convergence results (Theorem 3 and 4), how does the behavior policy influence the convergence?

---

> ### Author Response · Authors · 2024-11-21
> **Response to Reviewer nUhY**
>
> We appreciate the reviewer's feedback. We are glad that the reviewer found our model-free approach “new and interesting.” However, we believe that some misunderstandings regarding our work may have resulted in certain evaluations being overly harsh. We respectfully ask the reviewer to reconsider our paper in light of the clarifications provided below.
>
> ---
> >**W1**_“The experiments of the proposed methods are limited to simple synthesis setups.”_
>
> This paper extends standard off-policy evaluation algorithms to settings where Causal
> Consistency (Def. 2) does not hold, and unobserved confounding bias is generally present in the offline data. As far as we know, popular RL benchmarks do not explicitly model the challenges of unobserved confounding. Due to this reason, we have to build a synthetic evaluation environment in Windy Gridworld. Building more complex evaluation benchmarks modeling challenges of data bias is an exciting challenge and is currently under active development.
>
> ---
> >**W2** _“Lacking empirical comparisons with extensive body of partial identification OPE methods as mentioned in the related work section.”_
>
> Our work builds on discrete MDP with finite state-action domains while relaxing the key assumption of Causal Consistency (Def. 2), which allows the presence of unmeasured confounding and no-overlap. In other words, our algorithms do not impose additional constraints compared to the standard off-policy evaluation but only generalize it. On the other hand, as we explained in Appendix A, other confounding-robust OPE algorithms often invoke additional constraints, such as marginal sensitivity model or linearity, making them not applicable to our setting.
>
> ---
> >**Q1** _“How does the proposed algorithms perform on large-scale RL experiments?”_
>
> Our algorithms use tabular representations for value function estimation, which do not generalize well to large-scale, high-dimensional domains. Similar to standard OPE methods, it is possible to incorporate additional parametric forms about the value functions to overcome computational challenges. We acknowledge this is a promising direction and plan to explore it further.
>
> ---
> >**Q2** _“Despite that the true partial identification interval (defined through Theorem 1) gives valid upper and lower bounds of the true policy value, it seems that the resulting bound could be too optimistic or too pessimistic since the observational data can be induced by arbitrarily bad behavior policy (and sure this makes sense). Is it possible to provide further analysis how does the behavior policy influence the accuracy of the true partial identification interval? Or are there methods to avoid such kind of potential looseness under some circumstances?”_
>
> The quality of our bounds depends on the observed data, which is affected by the behavioral policy. There exist cases where our derived bounds in Theorems 1 and 2 are loose. However, they are also the tightest estimation given the input data and cannot be improved without additional assumptions. In those cases, it is reasonable to incorporate additional knowledge about the environment and behavioral policy to improve the evaluation.
>
> ---
> >**Q3** _“In addition to the asymptotic convergence results (Theorem 3 and 4), how does the behavior policy influence the convergence?”_
>
> The behavioral policy does not influence the convergence. This means that for cases where our derived bounds might be loose, the empirical estimation of these bounds is still efficient. This is different from applying importance sampling in standard OPE where reweighting the value function with the inverse propensity score could significantly increase the variance of the empirical estimators. In our proposed algorithms, the behavioral policy does not significantly affect the estimators’ variance.

---

> > ### Comment · Reviewer_nUhY · 2024-11-27
> >
> > Thank you very much for your detailed response to my concerns. It makes sense that the behavior policy might not influence the convergence, but could influence the tightness of the partial id bound, which, without further causal structural assumptions, is not improvable. I have raised my score to 6.

---

> > > ### Author Response · Authors · 2024-11-27
> > > **Thank you for your consideration**
> > >
> > > Thank you for taking the time to consider our feedback. We are glad to clarify the relationship between the behavior policy and the convergence aspects. If there are any further points for discussion, we would be happy to continue our conversation. Thanks again for your engagement.

---

### Official Review · Reviewer_eUhX · 2024-11-04

**Soundness:** 3
**Presentation:** 4
**Contribution:** 4
**Rating:** 8
**Confidence:** 4

**Summary:**

The paper studies off-policy evaluation in reinforcement learning when unobserved confounding exists in the data such that the causal consistency assumption is violated. Under this scenario, the paper derives causal Bellman equations to bound the value function and Q-function under a target policy. Two algorithms using eligibility traces are proposed to estimate the bounds of the value an Q-functions in both online and offline settings.

**Strengths:**

The paper is original in effectively integrating standard off-policy methods into bounding the value function in causal reinforcement learning, and the proposed algorithms seem straightforward to implement. It creatively addresses the causal inconsistency assumption that is present in many real-world applications. The problem formulation is clear. The properties of the proposed algorithms are backed with theoretical guarantee.

**Weaknesses:**

1. The environment considered in the paper is a little too restrictive, with finite action and state spaces, and bounded rewards.

2. The synthetic experiments are conducted in simple Windy Gridworld settings with a small action and state space.

**Questions:**

## Major comments:
1. The paper states that the proposed method relies on weaker assumptions than exisiting methods. In particular, the paper mentions that existing partial identification methods for off-policy evaluation rely on strong assumptions, including parametric assumptions about the system dynamics, model-based algorithm, and finite horizon. However, the settings consider in this paper actually rely on strong assumptions, including Markovness, finite action and state spaces, and bounded rewards. It would be great if the authors can review and compare other methods that consider the same settings, provided there are any.

2. The experiments are conducted in the simple synthetic Windy Gridworld environment. It would be helpful if the authors can comment on real-world scenarios to which the proposed methods are applicable, such as healthcare or robotics or something else. Experiments on real-world examples and comparisons with competing methods will further strengthen the paper.



## Minor comments:

1. Line 97 on page 2: "represents" --> "as"
2. Line 101 on page 2: Better to mention the full name before using the abbreviation "SCM"
3. Line 104 on page 2: Better to explicitly mention that $PA_V$ is the set of parents.
4. Line 361: "we" --> "We"

---

> ### Author Response · Authors · 2024-11-21
> **Response to Reviewer eUhX**
>
> We appreciate the reviewer's comments and feedback regarding typos and writing. We are pleased that the reviewer considered our work to be "original." Below, we address the main comments provided in the review.
>
> ---
> >**W1.** _"The environment considered in the paper is a little too restrictive, with finite action and state spaces, and bounded rewards. … However, the settings consider in this paper actually rely on strong assumptions, including Markovness, finite action and state spaces, and bounded rewards. It would be great if the authors can review and compare other methods that consider the same settings, provided there are any.”_
>
> First, we would like to clarify that standard off-policy evaluation algorithms such as temporal difference and eligibility trace rely on the assumptions of Markovness, finite action-state spaces, bounded rewards, and Causal consistency. Our work generalizes the critical assumption of Causal consistency (Def. 2), which precludes unobserved confounding bias in the offline data. We are able to, for the first time, extend the standard off-policy algorithms to more generalized settings where the target reward is not identifiable due to unobserved confounding and no-overlap. There exist algorithms in the RL literature (e.g., fitted Q-learning, deep Q-network) that generalize the finite action-state space assumption and extend off-policy evaluation to high-dimensional domains. We acknowledge that partial identification in confounded MDP with high-dimensional domains is a promising problem and is part of our future research plan.
>
> ---
> >**W2.** _“The synthetic experiments are conducted in simple Windy Gridworld settings with a small action and state space. … It would be helpful if the authors can comment on real-world scenarios to which the proposed methods are applicable, such as healthcare or robotics or something else. Experiments on real-world examples and comparisons with competing methods will further strengthen the paper.”_
>
> As far as we know, popular RL benchmarks do not explicitly model the challenges of unobserved confounding. Due to this reason, we have to build a customized evaluation environment in Windy Gridworld. There are many real-world scenarios where challenges of non-identifiability could arise. For example, learning from demonstration is a popular approach in robotics literature. The human demonstrator often has a different point of view and sensory capabilities from that of a learning robot. Consequently, some critical states observed by the human demonstrator were not captured by the robot’s input feature, leading to unobserved confounding bias. Deploying our proposed algorithm in these real-world examples is an exciting direction and is currently under active development.

---

### Official Review · Reviewer_Pp72 · 2024-11-12

**Soundness:** 3
**Presentation:** 3
**Contribution:** 3
**Rating:** 6
**Confidence:** 3

**Summary:**

The paper studies causal reinforcement learning (RL), i.e., RL in the presence of unmeasured confounding. The author(s) introduces a causal temporal difference (TD) learning and a causal eligibility traces algorithm for off-policy evaluation in causal RL, which combine TD or eligibility traces with the partial identification bounds developed in the econometrics or causal inference literature. Theoretically, causal Bellman equations were introduced to bound the Q- or value functions. Empirically, the author(s) also conducted numerical experiments to investigate the finite sample performance of their proposed algorithm.

**Strengths:**

As far as I can tell, the strengths of the paper include:

* The development of the proposed causal TD and causal eligibility traces algorithms, which appear novel;
* The proposed causal Bellman equations, which may provide practitioners with valuable insights into the value or Q-functions, especially in scenarios involving unmeasured confounders;
* The main text of the paper is technically sound. I did not review the Appendix, but I did not spot any errors in the main text;
* The writing is generally clear.

**Weaknesses:**

The paper suffers from two potential limitations:

* First, the numerical example is overly simplified. It only considers a 3 by 3 Windy Gridworld example. Additionally, the author(s) only reports the performance of their proposed algorithm, but did not compare their proposal against existing state-of-the-art methods. Given the huge literature on RL or/and off-policy evaluation (OPE) in the presence of unmeasured confounders, a comparison with these established methods would be highly beneficial. Such a comparison could highlight the most effective approaches for various applications. For instance, the methods developed by Kallus and Zhou (2020) and Bruns-Smith & Zhou (2023) seem directly relevant to addressing similar OPE challenges with unmeasured confounders. Additionally, POMDP-based methods, which use the POMDP framework to model the unmeasured confounding problem—such as those by Tennenholtz et al. (2020), Nair and Jiang (2021), and Shi et al. (2022) — would also be pertinent to this setting.

* Second, there is a lack of adequate discussions of the related literature. The last paragraph on Page 1 discusses the difference between this paper and other related works that use partial identification bounds. In particular, there is a line of work that "requires to additional parametric assumptions about the system dynamics". However, it would be better to detail this point later in the main text. What are the additional assumptions these papers imposed? How your proposal avoid imposing these assumptions? For instance, the paper by Namkoong (2020) needs a single-time unmeasured confounding assumption (I do not think this is a "parametric" assumption), which could be explicitly mentioned. Additionally, the paper by Bruns-Smith & Zhou (2023) also developed robust Bellman-operators using partial identification bounds. It would be better to clarify the difference between your proposal and theirs in detail. Moreover, the DGP mentioned on Page 2 suggests that you also relies on additional assumptions, more specifically, the memoryless unmeasured confounding assumption. It would be better to mention other related works that also rely on this assumption and discuss how to potentially verify this assumption in practice. Finally, as I mentioned, in addition to the use of partial identification methods, there are other methodologies, e.g., the POMDP-type methods to handle unmeasured confounding. These works are relevant as well.

**Questions:**

Refer to "Weaknesses"

---

> ### Author Response · Authors · 2024-11-21
> **Response to Reviewer Pp72 [1/2]**
>
> We thank the reviewer for the thoughtful feedback. We are glad the reviewer considered our problem “novel” and “technically sound.” We include additional discussion on related work in Appendix A. There is a growing body of research studying robust reinforcement learning under the presence of data bias and model uncertainties. Understandably, it is challenging to cover all of them. We will incorporate the reviewer’s suggestions and update the manuscript. We will address the main comments provided in the review below.
>
> ---
> >**W1.** _"First, the numerical example is overly simplified. It only considers a 3 by 3 Windy Gridworld example. Additionally, the author(s) only reports the performance of their proposed algorithm, but did not compare their proposal against existing state-of-the-art methods. Given the huge literature on RL or/and off-policy evaluation (OPE) in the presence of unmeasured confounders, a comparison with these established methods would be highly beneficial. Such a comparison could highlight the most effective approaches for various applications. For instance, the methods developed by Kallus and Zhou (2020) and Bruns-Smith & Zhou (2023) seem directly relevant to addressing similar OPE challenges with unmeasured confounders. Additionally, POMDP-based methods, which use the POMDP framework to model the unmeasured confounding problem—such as those by Tennenholtz et al. (2020), Nair and Jiang (2021), and Shi et al. (2022) — would also be pertinent to this setting.”_
>
> This paper extends standard temporal difference (TD) algorithms in Markov decision processes (MDP) to account for unobserved confounding bias and no-overlap in the offline data. Similar to standard TD algorithms, we focus on discrete MDP with finite state-action domains. The main difference is that we relax the Causal Consistency (Def. 2) which allows the policy effect to be identifiable in standard off-policy evaluation. As far as we are aware, this setting is nove, and we propose the first model-free, TD algorithms using eligibility traces for confounding-robust off-policy evaluation.
>
> More specifically, the methods developed by Kallus and Zhou (2020) and Bruns-Smith & Zhou (2023) rely on the assumption called the marginal sensitivity model. This line of work utilizes a bound over the odds ratio of the nominal and the actual behavioral policies, which is not accessible in our setting. Similarly, POMDP-based methods require additional parametric knowledge on the latent exogenous variables, e.g., discreteness or weakly-revealing property, which is assumed to be unknown in our setting.
>
> Our proposed methods apply to MDP environments with finite state-action space and bounded rewards, which are verifiable from observed data. We acknowledge that extending the proposed algorithms to confounded MDP with high-dimensional domains is an exciting challenge and is part of our future research plan.

---

> ### Author Response · Authors · 2024-11-21
> **Response to Reviewer Pp72 [2/2]**
>
> ---
> >**W2.** _“Second, there is a lack of adequate discussions of the related literature. The last paragraph on Page 1 discusses the difference between this paper and other related works that use partial identification bounds. In particular, there is a line of work that "requires to additional parametric assumptions about the system dynamics". However, it would be better to detail this point later in the main text. What are the additional assumptions these papers imposed? How your proposal avoid imposing these assumptions? For instance, the paper by Namkoong (2020) needs a single-time unmeasured confounding assumption (I do not think this is a "parametric" assumption), which could be explicitly mentioned. Additionally, the paper by Bruns-Smith & Zhou (2023) also developed robust Bellman-operators using partial identification bounds. It would be better to clarify the difference between your proposal and theirs in detail. Moreover, the DGP mentioned on Page 2 suggests that you also relies on additional assumptions, more specifically, the memoryless unmeasured confounding assumption. It would be better to mention other related works that also rely on this assumption and discuss how to potentially verify this assumption in practice. Finally, as I mentioned, in addition to the use of partial identification methods, there are other methodologies, e.g., the POMDP-type methods to handle unmeasured confounding. These works are relevant as well.”_
>
> First, to clarify, our methods build on the settings of standard temporal difference and eligibility trace algorithms: the underlying environment is a Markov decision process with finite state-action space. We account for the same standard assumptions while generalizing Causal Consistency (Def. 2), allowing the violations of no unmeasured confounding and overlap. Consequently, our algorithms do not impose additional constraints compared to the standard off-policy evaluation but only generalize it.
>
> There is a line of work in confounding robust off-policy evaluation, e.g., Zhou and Kallus (2018), Namkoong (2020), and Bruns-Smith & Zhou (2023), that utilizes an assumption called the marginal sensitivity model, which bounds the odds ratio between the nominal and real behavioral policies. We categorize this assumption as parametric since it requires some smoothness in the functional relationships between the unobserved confounder $U$ and the treatment $X$, precluding certain parametric forms of behavioral policies. However, we understand some readers might find this categorization confusing. We have updated the manuscript to clarify this comment.
>
> Finally, this work focuses on off-policy evaluation in MDPs where the influence of observed state and unobserved confounders are often local. Off-policy evaluation in POMDP requires detailed parametric constraints on the latent state or the target policy, which is beyond the scope of this paper. We will add additional discussion in Appendix A for POMDP-based methods.

---

### Comment · Reviewer_eUhX · 2024-11-23

I would like to thank the authors for the detailed responses and comments. I think the paper targets an important issue, unconfoundedness, in causal learning. The authors' responses are reasonable. I would like to maintain my positive rating for the paper.

---

### Meta-Review · Area_Chair_9BsS · 2024-12-23

**Metareview:**

The paper studies off-policy evaluation from biased offline data where target and behavior policy do not share a common support and where the bias due to unobserved confounding cannot be ruled out a priori.
The reviewers praise the novelty of the causal temporal difference and eligibility traces methods, the analysis with the causal Bellman equations, and the writing of the paper.
Potential limitation, highlighted by all reviewers, include the fact that the experiments operate in a setting that is too simplistic, and there seems to be a lack of discussion of important related literature.

**Additional Comments On Reviewer Discussion:**

Some of the structural weaknesses have not been addressed

---

### Decision · Program_Chairs · 2025-01-22

Reject